# ACTIVE LEARNING FOR OBJECT DETECTION WITH EVIDENTIAL DEEP LEARNING AND HIERARCHICAL UNCERTAINTY AGGREGATION

**Younghyun Park**[1]**, Wonjeong Choi**[1]**, Soyeong Kim**[1]**, Dong-Jun Han**[2*]**, Jaekyun Moon**[1]
[1] Korea Advanced Institute of Science and Technology (KAIST), [2] Purdue University
{dnffkf369,dnjswjd5457,best004}@kaist.ac.kr, han762@purdue.edu, jmoon@kaist.edu

## ABSTRACT

Despite the huge success of object detection, the training process still requires an immense amount of labeled data. Although various active learning solutions for object detection have been proposed, most existing works do not take advantage of epistemic uncertainty, which is an important metric for capturing the usefulness of a sample. Also, previous works pay little attention to the attributes of each bounding box (e.g., nearest object, box size) when computing the informativeness of an image. In this paper, we propose a new active learning strategy for object detection that overcomes the shortcomings of prior works. To make use of epistemic uncertainty, we adopt evidential deep learning (EDL) and propose a new module termed model evidence head (MEH), that makes EDL highly compatible with object detection. Based on the computed epistemic uncertainty of each bounding box, we propose hierarchical uncertainty aggregation (HUA) for obtaining the informativeness of an image. HUA realigns all bounding boxes into multiple levels based on the attributes and aggregates uncertainties in a bottom-up order, to effectively capture the context within the image. Experimental results show that our solution outperforms existing state-of-the-art methods by a considerable margin.

## 1 INTRODUCTION

Deep learning contributes to huge success in computer vision problems such as semantic segmentation (Long et al., 2015; Ronneberger et al., 2015; Chen et al., 2018) and object detection (Liu et al., 2016; Lin et al., 2017; Redmon et al., 2016). However, training a deep neural network typically comes with a cost of large labeled datasets. Labeling data for complex vision problems requires intensive labor of human experts, which makes preparing for practical application challenging. Active learning, which gradually labels a set of samples based on the informativeness (e.g., uncertainty), is a promising solution for this problem due to its simplicity and high performance.

Although active learning has been extensively studied on image classification, only a few prior works focused on object detection (Yuan et al., 2021; Su et al., 2020; Haussmann et al., 2020; Yu et al., 2021) despite its practical importance. Furthermore, existing works on active learning for object detection have two limitations. First, when computing the informativeness of an image, most previous works only use the *aleatoric uncertainty*, not taking the *epistemic uncertainty* into account. Epistemic uncertainty, also known as knowledge uncertainty, captures the lack of knowledge of a model (caused by a lack of data) and can be reduced when large amounts of data are available. Aleatoric uncertainty, on the other hand, captures the noise inherent in the observed data and is irreducible. As stated in (Nguyen et al., 2022; Hafner et al., 2018; Hüllermeier & Waegeman, 2021), epistemic uncertainty can reflect the usefulness of samples and support active learning better than aleatoric uncertainty. Secondly, previous works on active learning for object detection generally ignore the attributes of bounding boxes (e.g., nearest object, box size) when computing the informativeness of an image: informativeness is often defined as the maximum or mean of the uncertainty values of all bounding boxes in the image. This can be a problem because a cluttered image with many objects belonging to various categories can be enforced to have a similar uncertainty value relative to just a simple image with only a few objects belonging to a single category.

---

[*]Corresponding author

**Goal and challenge.** The general goal of this paper is to propose an active learning strategy for object detection, that handles the above limitations of existing works. First, we aim to build an algorithm that can compute epistemic uncertainty quickly yet correctly in object detection. Some existing works on object detection (Haussmann et al., 2020; Feng et al., 2019) calculate epistemic uncertainty using multi-model based methods (e.g., model ensemble (Beluch et al., 2018), Monte Carlo (MC) dropout (Gal & Ghahramani, 2016)). However, these methods require multiple models or repetitive forward propagations for MC integration, consequently making practical applications difficult. Secondly, we aim to design an uncertainty aggregation scheme which can consider attributes of bounding boxes and understand the context within images. This goal is meaningful since aggregation schemes of previous works (Yuan et al., 2021; Roy et al., 2018; Choi et al., 2021), which simply rely on the maximum/mean of all bounding boxes, are hard to reflect the context in images.

**Main contributions.** Our first key idea is to adopt Evidential Deep Learning (EDL) to effectively compute epistemic uncertainty, and to propose a new module that makes EDL highly compatible with object detection. Introduced by (Sensoy et al., 2018; Amini et al., 2020), EDL is a useful tool to compute epistemic uncertainty for detecting unfamiliar data (e.g., unseen unlabeled data) since it samples model ensembles almost instantly. However, previous works on EDL have mainly focused on image classification and induce unconfident prediction and unstable training when simply applied to object detection. To this end, we propose a new module named as Model Evidence Head (MEH) to enable confident prediction and stable training of EDL. Specifically, MEH predicts the expected difficulty, or model evidence, and is optimized independently of the object detector. To our knowledge, this is the first research to make EDL compatible with object detection on 2D images.

Another key ingredient for our solution is Hierarchical Uncertainty Aggregation (HUA), which makes use of attributes in bounding boxes for computing the informativeness of an image. According to the attributes, HUA realigns boxes into multiple levels and aggregates uncertainties in a bottom-up order. This helps to capture the context within the image and improves the quality of the expected informativeness of images. Overall, our main contributions are summarized as follows:

- We make use of EDL to effectively compute epistemic uncertainty in object detection, and design a new module termed **Model Evidence Head (MEH)** which solely predicts the model evidence independently of the class confidence to make EDL adaptable to object detection.
- We propose **Hierarchical Uncertainty Aggregation (HUA)**, which reorganizes all bounding boxes into several levels and aggregate uncertainties of each level in a bottom-up manner, to better capture the context within the image.

We validate the efficacy of proposed methods using RetinaNet and SSD as base models on well-known datasets: PASCAL VOC, MS-COCO. Extensive experiments demonstrate that the proposed methods significantly improve performance achieving new state-of-the-art results.

## 2 RELATED WORKS

**Active learning for object detection.** Active learning (Sinha et al., 2019; Yoo & Kweon, 2019; Sener & Savarese, 2017; Gal et al., 2017; Wang et al., 2016) aims to select a small subset of informative unlabeled samples which is expected to be most effective. Although active learning has been extensively studied for the classification problem, only a few works focus on object detection (Yuan et al., 2021; Su et al., 2020; Haussmann et al., 2020; Yu et al., 2021), despite its practical importance. (Yuan et al., 2021; Su et al., 2020) train discriminators using unlabeled data to predict whether an image is from the labeled set or the unlabeled set. LL4AL (Yoo & Kweon, 2019) trains an auxiliary module to predict loss of data samples, where a sample with high predicted loss is considered as the one with high informativeness. The concurrent work CDAL (Agarwal et al., 2020) introduced a distance measure to select diverse samples in semantic and spatial context. Recently, (Choi et al., 2021) predict the parameters of Gaussian mixture models and computes epistemic uncertainty as the variance of Gaussian modes. However, there is no guarantee that model uncertainty leads to variance in GMM, and the size of the model ensemble is fixed to be small since the number of Gaussian modes is not allowed to change after training. Overall, existing approaches do not fully take advantage of epistemic uncertainty and pay little attention to attributes of bounding boxes during uncertainty aggregation. Our work resolves both limitations using novel and effective methods.

**Bayesian deep learning.** The goal of Bayesian deep learning is to build a credible machine that measures uncertainty on its decision as well. In contrast to the frequentist approach where model

parameters are not random variables but fixed quantities, the Bayesian approach assumes prior distribution on model parameters and estimates posterior distributions based on given data. Recently, many computer vision works have proposed the use of uncertainty through Bayesian deep learning. (Kendall & Gal, 2017) investigates the benefits of modeling epistemic/aleatoric uncertainty in vision tasks, while (Simon et al., 2022) and (Guo et al., 2022) use the reparameterization trick to estimate uncertainty for neural architecture search and action recognition, respectively. However, previous Bayesian deep learning methods such as Bayes by Backprop (Gal & Ghahramani, 2015; Blundell et al., 2015), MC dropout (Gal & Ghahramani, 2016) and Variational Inference (Blei et al., 2017; Hoffman et al., 2013; Kingma & Welling, 2013) often require costly sampling-based MC approximation to estimate the posterior distribution. Instead, our work adopts Evidential Deep Learning (EDL) (Sensoy et al., 2018; Amini et al., 2020) to compute epistemic uncertainty through a single forward propagation. Different from typical Bayesian approaches, EDL tries to predict prior distributions of posterior models without repetitive model sampling. However, EDL has hardly been applied to complicated computer vision problems such as object detection yet, and it usually requires adversarial learning using a regularization loss which makes training unstable. In this work, we propose MEH to overcome aforementioned limitations of previous works.

## 3 PROPOSED METHOD

### 3.1 PROBLEM SETUP AND ORGANIZATION

Active learning consists of multiple learning cycles. At the first cycle, a large unlabeled dataset $U^0$ and a small labeled dataset $L^0$ are given. Once an object detector is trained using $L^0$, the network selects the most informative images $I^0$ from $U^0$, based on some measures. Human oracles then manually label $I^0$ and construct new labeled/unlabeled sets for the next cycle as $L^1 = L^0 \cup I^0$ and $U^1 = U^0 \setminus I^0$. The same process is repeated until the annotation budget is exhausted.

In this section, we describe our method for computing the informativeness of an image for active learning in object detection. Section 3.2 describes how we estimate epistemic uncertainty of a bounding box using EDL and the proposed MEH. Based on the results in Section 3.2, we describe how we compute the informativeness of an image using HUA, in Section 3.3.

### 3.2 EVIDENTIAL DEEP LEARNING FOR EPISTEMIC UNCERTAINTY

**Overview of our approach.** Fig. 1 shows the high-level descriptions of our approach under the EDL framework for computing epistemic uncertainty of a bounding box. MEH in Fig. 1a is our new module that makes EDL adabtable to object detection. Under the Dirichlet-Categorical Bayesian framework, we propose to predict class confidences $\beta = \{\beta_k\}_{k=1}^K$ and model evidence $\lambda$, to obtain the concentration parameter $\alpha = \{\alpha_k\}_{k=1}^K$ and construct the prior Dirichlet distribution $Dir(\theta|\alpha)$ for each bounding box. Epistemic uncertainty is then computed using model ensembles $\theta \sim Dir(\theta|\alpha)$ as the mutual information between predictions and model posterior as

$$\underbrace{\mathcal{I}[y,\theta]}_{\text{Epistemic Unc. } (U_{epi})} = \underbrace{\mathcal{H}\big[\mathbb{E}_{p(\theta|\alpha)}[p(y|\theta)]\big]}_{\text{Total Unc. } (U_{tot})} - \underbrace{\mathbb{E}_{p(\theta|\alpha)}\big[\mathcal{H}[p(y|\theta)]\big]}_{\text{Aleatoric Unc. } (U_{ale})}, \tag{1}$$

where $\mathcal{H}$ denotes Shannon entropy and $\theta$ parameterizes categorical likelihood $Cat(\theta)$; $p(y|\theta)$ and $p(\theta|\alpha)$ are probability functions of categorical and Dirichlet distributions. Fig. 1b shows an example on how the 3 types of uncertainty in (1) change according to $\lambda$ predicted from our MEH module.

Prior works on active learning without model sampling fail to take advantage of epistemic uncertainty $U_{epi}$ since $p(\theta|\alpha)$ becomes a point estimate, enforcing $U_{tot} = U_{ale}$ and $U_{epi} = 0$. Our two-stage Bayesian framework based on EDL enables us to sample $\theta$ almost instantly, which is significantly faster than multi-model-based Bayesian methods such as MC-dropout (Gal & Ghahramani, 2016) and model ensemble (Haussmann et al., 2020). In the following, we start by applying EDL to object detection to describe our algorithm in details.

**Applying EDL to object detection.** Inspired by EDL (Sensoy et al., 2018; Amini et al., 2020; Zhao et al., 2020), unlike typical object detectors where the classification head predicts parameters $\theta$ of the categorical distribution, we predict a high-order Dirichlet distribution $Dir(\theta|\alpha)$ which is a conjugate prior of the lower-order categorical likelihood $Cat(\theta)$. To fit our evidential model to data, we maximize the marginal likelihood, also known as maximum likelihood Type II. The marginal

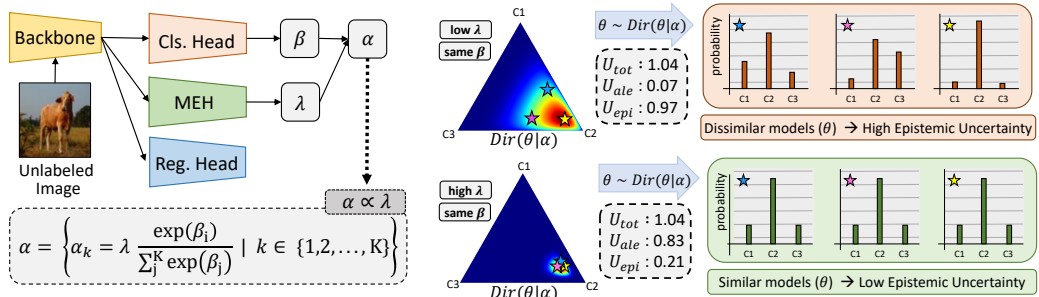

(a) Obtaining concentration parameter $\alpha$      (b) Calculation of epistemic uncertainty with sampling

Figure 1: Proposed EDL-based uncertainty computation of a bounding box. (a) First, for an unlabeled image, the classification head produces class confidences $\beta = \{\beta_i\}_{i=1}^K$ while our model evidence head (MEH) produces model evidence $\lambda$. $\beta$ and $\lambda$ are used to compute a parameter set $\alpha$ of Dirichlet distribution $Dir(\theta|\alpha)$. (b) Based on $\alpha$, parameters $\theta$ of categorical distribution $Cat(\theta)$ are sampled from $Dir(\theta|\alpha)$. Epistemic uncertainty is then computed as a dissonance between sampled $\theta$. Note that a larger $\lambda$ indicates a larger $\alpha$, making $Dir(\theta|\alpha)$ sharper; sharp $Dir(\theta|\alpha)$ produces similar $Cat(\theta)$, decreasing the epistemic uncertainty.

likelihood for a bounding box $y_i$ can be obtained by marginalizing over the likelihood parameter $\theta$:

$$p(y_i|\alpha) = \int p(y_i|\theta)p(\theta|\alpha)d\theta = \int \prod_{j=1}^{K} p_j^{\mathbb{1}\{y_j=i\}} \frac{\Gamma(\sum_{j=1}^{K}\alpha_j)}{\prod_{j=1}^{K}\Gamma(\alpha_j)} \prod_{j=1}^{K} p_j^{\alpha_j-1} dp. \tag{2}$$

The first term in the integral comes from the categorical distribution, while the second term is from the Dirichlet distribution; the integration can be interpreted as marginalization over every possible categorical model $\theta \sim Dir(\theta|\alpha)$. Thanks to the Dirichlet-Categorical conjugacy, the integral can be written in a closed-form:

$$p(y_i|\alpha) = \frac{\Gamma(\sum_{j=1}^{K}\alpha_j)}{\prod_{j=1}^{K}\Gamma(\alpha_j)} \cdot \frac{\prod_{j\neq i}^{K}\Gamma(\alpha_j)\Gamma(\alpha_i+1)}{\Gamma(\sum_{j=1}^{K}\alpha_j+1)} = \frac{\alpha_i}{\sum_j \alpha_j}. \tag{3}$$

The network is then optimized to minimize negative log marginal likelihood $L_{cls} = -\sum_k \bar{y}_k \log(p(y=k|\alpha))$, where $\bar{y}$ is an one-hot label vector. At inference, the expected probability for the $k$-th category is computed as $\hat{p}_k = \alpha_k/S$, where $S = \sum_k \alpha_k$ is the Dirichlet strength.

**Issues.** When the EDL method used for image classification is directly applied to object detection, several problems arise: (i) training becomes unstable due to the adversarial regularization loss, and (ii) the prediction becomes unconfident due to the use of ReLU as a hypothesis function. Given class confidences $\beta = \{\beta_i\}_{i=1}^K$, previous works (Sensoy et al., 2018; Zhao et al., 2020; Hemmer et al., 2022) adopt ReLU to compute the concentration parameter as $\alpha'_k = \text{ReLU}(\beta_k)+1$. Although ReLU performs well in image classification with simple datasets, it noticeably decreases the mAP score in object detection where the number of classes is large and confident prediction is important. For example, in the case of 80-way classification, $\beta_k$ should be at least 7820 to achieve $\hat{p}_k = 0.99$ even when $\beta_i = 0$ for all $i \neq k$, according to $\alpha'_k = \text{ReLU}(\beta_k) + 1$ and (3).

**MEH module.** To overcome the aforementioned issues, we propose MEH and introduce several techniques. As a first step, we apply softmax as $\alpha_k = \frac{\exp(\beta_k)}{\sum_c \exp(\beta_c)}$ instead of ReLU to produce sufficiently confident prediction. However, although softmax enables confident prediction, it forces the Dirichlet strength $S = \sum_k \alpha_k$ to be 1. This makes the entropy of Dirichlet distribution unchangeable and deprives the model of the ability to predict model evidence. To tackle this issue, we propose a new module termed MEH, which solely predicts model evidence $\lambda$. When $\lambda$ is obtained from MEH and class confidences $\{\beta_k\}_{k=1}^K$ are obtained from the classification head, we propose to re-scale the concentration parameter $\alpha$ as

$$\alpha_k = \lambda \frac{\exp(\beta_k)}{\sum_c \exp(\beta_c)}, \tag{4}$$

where $\beta_c$ is the class confidence for class $c$. The above re-scaling process with $\lambda$ transforms $Dir(\theta|\alpha)$ to be either concentrated or flat. Epistemic uncertainty is then computed based on the re-scaled $Dir(\theta|\alpha)$ via (1). Fig.1b illustrates the effect of $\lambda$ on epistemic uncertainty. The effects of softmax and $\lambda$ are validated via experiments in Section 4.

**MEH loss function.** We separate the training of MEH from the primary object detector with a novel loss function for stability. For a Bayesian model to work properly, it must be able to recognize the "I don't know" state, i.e., the model should generate low model evidence for difficult samples. To this end, previous works on EDL (Hemmer et al., 2022; Sensoy et al., 2018) shrink the model evidence for singletons which do not correspond to target category, via the following regularization loss

$$L_{reg}(\alpha, y) = KL[Dir(\theta|\hat{\alpha})||Dir(\theta| < 1, ..., 1 >)], \tag{5}$$

where $\hat{\alpha} = \bar{y} + (\mathbb{1} - \bar{y}) \odot \alpha$ is the concentration parameter where the element for target category is disregarded. However, this loss enforces the non-target elements to be one, no matter how large an error is detected, since the uniform Dirichlet distribution $Dir(\theta| < 1, ..., 1 >)$ is considered as a constant supervisory signal. Furthermore, this adversarial regularization loss suppresses the overall model evidence and makes training unstable. To overcome these issues, we propose a simple yet effective solution: we utilize the continuous loss score from the classification head as a target value. Specifically, when classification loss $l_{s,i}$ is given for bounding box $i$ at scale $s$, the proposed loss function for MEH is defined as

$$L_{MEH}(\lambda, l) = \sum_s \sum_i \left(l_{s,i} - \frac{1}{\lambda_{s,i}}\right)^2. \tag{6}$$

This loss encourages MEH to decrease model evidence $\lambda_{s,i}$ when the classfication loss $l_{s,i}$ is large; similarly, MEH is guided to increase $\lambda_{s,i}$ for small $l_{s,i}$. $L_{MEH}$ has the same effect as the previous regularization loss (5) in that it prevents overconfidence and helps the model to predict uncertainty better; however, by using the target loss as a changeable supervisory signal, $L_{MEH}$ is able to dynamically penalize the model to better predict the model evidence.

**Stability.** Note that we deliberately supervise only model evidence $\lambda$ independently of Dirichlet parameters $\{\alpha_k\}_{k=1}^K$ to train MEH. $\lambda$ is utilized only for re-scaling the Dirichlet strength and $\lambda$ has no effect on $p(x|\theta)$ or $\hat{p}_k$, since $\lambda$ will naturally vanish when division $\alpha_k/S$ occurs. Hence, the MEH network $\Phi_{MEH}$ and remaining network $\Phi \setminus \Phi_{MEH}$ can be updated in a disjoint manner. We in turn optimize $\Phi_{MEH}$ and $\Phi \setminus \Phi_{MEH}$, thus training of $\Phi_{MEH}$ never affects the performance of the primary object detector; this resolves the stability issue of previous EDL approaches.

**Applications to recent models.** Our method is not limited to softmax-based object detectors like SSD (Liu et al., 2016), but also applicable to sigmoid-based detectors like RetinaNet (Lin et al., 2017). Given sigmoid-based prediction $p \in [0, 1]$, focal loss can be defined as $FL(p_t) = -(1 - p_t)^\gamma \log(p_t)$ where $p_t = p$ if $y = 1$, $p_t = 1 - p$ otherwise. To enable evidential learning, we replace sigmoid-based binary prediction $p$ by Dirichlet-Categorical multiclass prediction $p(x|\alpha)$ of equation (3). We train RetinaNet with the modified evidential focal loss and report its performance in Fig. 3.

## 3.3 HIERARCHICAL UNCERTAINTY AGGREGATION

Based on the epistemic uncertainty of a bounding box, we propose HUA: a new method for computing the final informativeness score of an image. Recent active learning methods for object detection (Yuan et al., 2021; Choi et al., 2021; Yu et al., 2021) typically compute the informativeness of an image as the mean or maximum of all bounding boxes. To this end, we propose a generalized framework that can apply different kinds of aggregation functions based on the level of information (predicted category, box size, nearby object) of each bounding box, as described in Fig. 2.

**Filtering bounding boxes.** Single-stage object detectors such as RetinaNet and SSD generate bounding boxes at every scale, pixel and anchor. Since most boxes correspond to background, we first filter out background boxes whose $\max_k(\hat{p}_k)$ is lower than threshold $\gamma_{score}$.

**Realigning bounding boxes.** Besides the uncertainty score, each bounding box contains much more information: predicted object, scale and category it belongs to. Based on these information, we propose to realign the boxes into a hierarchical structure. First, boxes are matched to the nearest object depending on the IoU score. For example, in Fig. 2a, the blue box is matched to "object", but the green box is ignored since IoU is lower than the threshold $\gamma_{IoU}$. Secondly, boxes are further grouped based on the scale to which they belong. Lastly, boxes are divided based on the category ($\arg\max_k \hat{p}_k$) to which they belong. See Fig. 2b for the resulting hierarchy of bounding boxes.

**Uncertainty aggregation.** Once the bounding boxes are fully realigned, individual uncertainty scores are unified in a hierarchical order. As shown in Fig. 2b, uncertainties of the boxes in a lower level (e.g., class) are aggregated into a single value through a predefined aggregation function;

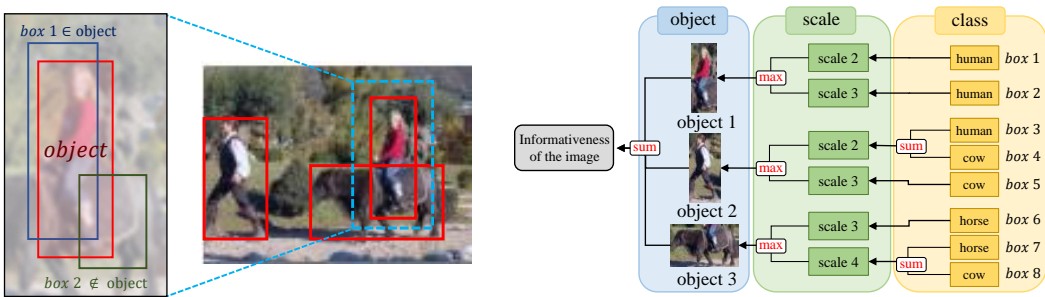

(a) Matching of bounding boxes          (b) Uncertainty aggregation based on attributes

Figure 2: An overview of the proposed hierarchical uncertainty aggregation (HUA). (a) Bounding boxes are grouped or ignored based on IoU with the near objects. (b) Bounding boxes in an image are hierarchically realigned based on object, scale, category to which the boxes belong. Uncertainties of bounding boxes in the same level are aggregated and then passed to the higher level. For the above example, a set of functions (sum, max, sum) is chosen for each information level (object, scale, class).

the aggregated value is then passed to an upper level (e.g, scale). This aggregation repeats from the "class level" to "object level". Note that HUA can be viewed as a generalized scheme where different types of aggregation functions can be adopted at different levels of information; it can cover aggregation methods of previous works (Choi et al., 2021; Yuan et al., 2021) where identical functions (max or mean) are applied to the every level. As discussed in Appendix, we specifically propose to adopt *sum* operation when aggregating the uncertainties at the object level, since this reflects the number of objects in final informativeness of the image. We also propose to apply *sum* operation to the class level. Given that ambiguous bounding boxes are often classified into more than one class, considering these classes altogether helps to increase the uncertainty score of the difficult object. By considering the attributes of each bounding box, HUA effectively captures the context within the image in the informativeness score.

### 3.4 SELECTION OF INFORMATIVE IMAGE

Now the epistemic uncertainties of all unlabeled images can be computed at each active learning cycle. While previous works typically select the top-$k$ uncertain images, we propose to select from the filtered-out images as well. These images are also valuable since the model was incapable of sensing any objects due to lack of knowledge. Composing a certain fraction of selections with these images further improves the performance, where the detailed analysis is provided in Appendix.

## 4 EXPERIMENTS

### 4.1 EXPERIMENTAL SETUP

**Dataset.** Our work is validated on PASCAL VOC (Everingham et al., 2010) and MS-COCO (Lin et al., 2014). As to PASCAL VOC, VOC07+12 *trainval* is used for training and VOC07 *test* is used for evaluation. While mAP50 is used for evaluation metric in PASCAL VOC, mean of mAP at IOU=.50:.05:.95 is used for MS-COCO. In the first cycle for PASCAL VOC, 5% from 16,551 samples are randomly selected. Then 2.5% of samples are labeled at each cycle until reaching 20%. As for MS-COCO with 117,267 samples, labeled sets increased from 2% to 10% in steps of 2%.

**Implementation details.** For fair comparisons with previous works (Yuan et al., 2021; Agarwal et al., 2020; Yoo & Kweon, 2019), we adopt RetinaNet (Lin et al., 2017) and SSD (Liu et al., 2016) which use ResNet50 and VGG16 as backbones, respectively. The structure of MEH is the same as the regression head except that the dimension of the output layer is one. As for RetinaNet, we follow the setting of (Yuan et al., 2021) and set epoch to be 26, batch size to be 2; also, we follow the setting of (Yoo & Kweon, 2019; Agarwal et al., 2020) for SSD and set epoch to be 300, batch size to be 32. In both experiments on PASCAL VOC and MS-COCO, the SGD optimizer is used with a learning rate of 0.001; a $\ell_2$ weight decay rate of 0.0001; a momentum of 0.9; and a warm-up strategy for the early 500 steps. As for the backbones, ResNet50 and VGG16 are pretrained on ImageNet. Note that at every cycle, the object detector is trained from scratch with the pretrained backbone. We implement our framework using Pytorch 1.5.0 and mmdetection 2.13.0. In every experiment, single NVIDIA GeForce RTX 2080Ti is used. We set the confidence threshold for box

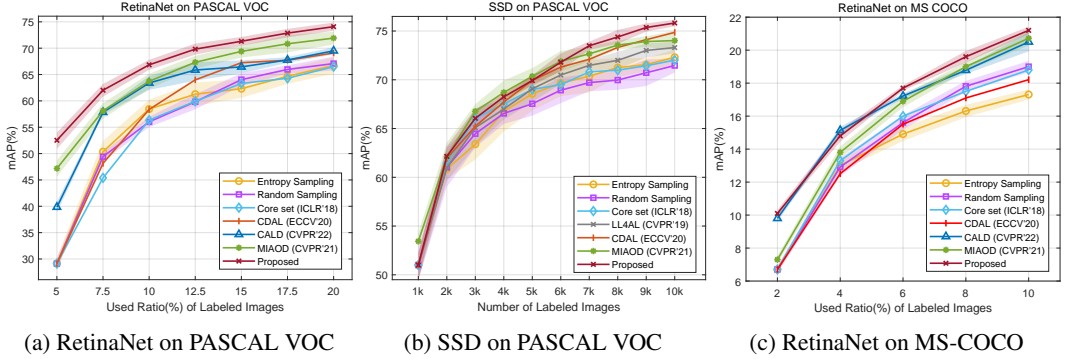

Figure 3: Performance comparison with state-of-the-art methods.

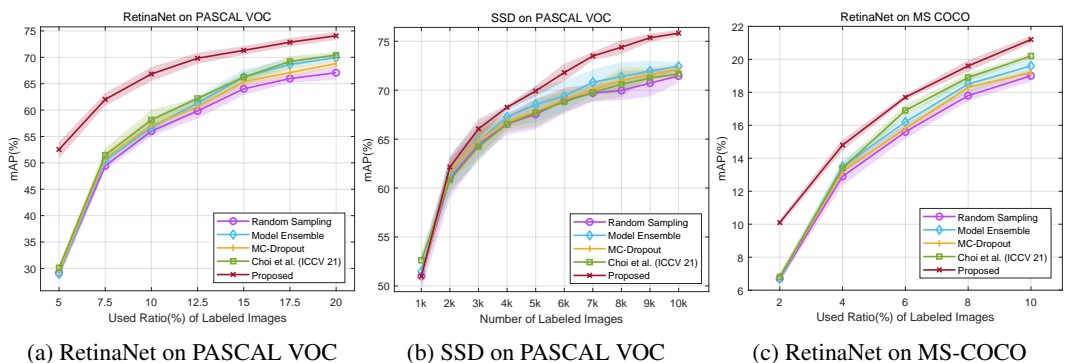

Figure 4: Performance comparison with multi-model-based Bayesian methods.

filtering $\gamma_{score}$ to 0.3, and IoU threshold for object grouping $\gamma_{IoU}$ to 0.5. 500 ensemble members $\theta$ are sampled for uncertainty estimation. For our scheme, (sum, max, sum) operations are adopted for each information level (object, scale, class) when applying HUA. Ablation studies with different operations are provided in Appendix. We compose 15% of selections with filtered images, where the results with various selection ratios are reported in Appendix.

**Baselines.** We compare our work with state-of-the-art works: MI-AOD (Yuan et al., 2021), CDAL (Agarwal et al., 2020), LL4AL (Yoo & Kweon, 2019), Core-set (Sener & Savarese, 2017), CALD (Yu et al., 2022). Also, as basic baselines, *Entropy Sampling* (Roy et al., 2018), *Random Sampling* with vanilla RetinaNet/SSD are considered as well. For Core-set, features from backbone (ResNet50/VGG16) are used following the setting of (Yoo & Kweon, 2019; Choi et al., 2021). While *Random Sampling* randomly selects unlabeled samples to be labeled for the next cycle, *Entropy Sampling* selects the samples based on Shannon entropy; the average value of Shannon entropy among all of bounding boxes is regarded as the informativeness as done in (Yuan et al., 2021; Choi et al., 2021; Roy et al., 2018). Five independent networks are trained with different seeds and the averaged performance is reported with 95% confidence intervals.

## 4.2 MAIN EXPERIMENTAL RESULTS

**Comparison with state-of-the-arts.** Fig. 3 shows mAP scores under various settings. Our proposed method exhibits a clear edge over other baselines, giving the best performance. In Figs. 3a and 3c, the proposed method shows superiority in early cycles, proving that the proposed evidential focal loss is effective when labeled data is extremely limited. It is noteworthy to consider our method is trained only on a labeled set, while MI-AOD benefits from semi-supervised learning using an unlabeled set as well. In Fig. 3b, the reason SSD has lower initial performance than RetinaNet comes from the difference of the loss function. Unlike RetinaNet trained with the focal loss, SSD is trained with the CE loss. As in equation (3), since we calculated $\alpha$ using a softmax, the loss expression does not change at all in the case of SSD. Still, it is remarkable that our methods (MEH, HUA) gradually increase the performance by selecting informative images even with the same loss.

**Comparison with multi-model based Bayesian methods.** In recent active learning works, multi-model based methods are used to compute epistemic uncertainty. Here, we compare existing meth-

ods and our work. As for baselines, we first consider MC-dropout; following (Beluch et al., 2018; Choi et al., 2021), dropout layers whose rate is 0.1 are added to the convolutional layers and 25 forward passes are sampled to compute epistemic uncertainty using equation (1). Secondly, model ensemble is considered, where we follow (Haussmann et al., 2020; Choi et al., 2021) and train three independent models to construct an ensemble. Lastly, we consider the work of (Choi et al., 2021); it predicts the parameter of a Gaussian mixture model (GMM) and regards the variance of Gaussian modes as the epistemic uncertainty. In every baseline, we average uncertainties from all bounding boxes for the computation of the final image-level uncertainty, as done in (Yuan et al., 2021; Choi et al., 2021; Roy et al., 2018). In Fig. 4, we first observe that multi-model based methods yield higher scores than random sampling, underlying the importance of epistemic uncertainty. Still, the large performance gap between the baselines and our work confirms the benefits of the proposed techniques: MEH and HUA. It is noteworthy that baselines ignore the hierarchy between bounding boxes, which implies that there possibly exist important yet underestimated boxes.

### 4.3 ABLATION STUDIES AND DISCUSSIONS

**Effects of MEH and HUA.** Table 7 shows ablation studies on the proposed methods. While *Random* and *Entropy* use vanilla RetinaNet with focal loss, proposed methods use the proposed evidential focal loss. While *ReLU* computes $\alpha$ with ReLU as done in (Sensoy et al., 2018; Zhao et al., 2020; Hemmer et al., 2022), *Softmax* uses softmax. All methods except *HUA* compute uncertainty of an image as the mean of all boxes, as in (Yuan et al., 2021; Choi et al., 2021). At every cycle, the proposed methods (Softmax, MEH, HUA) increase performances by a large margin. The results indicates that informativeness can be captured much better via our proposed methods. Interestingly, it can be seen that the *ReLU* method performs even worse than *Random*. This result confirms that previous works of EDL for image classification cannot be simply applied to object detection.

| Method | Ratio (%) of Labeled Samples | | | | | | |
|---|---|---|---|---|---|---|---|
| | 5.0 | 7.5 | 10.0 | 12.5 | 15.0 | 17.5 | 20.0 |
| Random | 29.13 | 50.35 | 58.45 | 61.27 | 62.31 | 64.67 | 66.72 |
| Entropy (Roy et al., 2018) | 29.13 | 49.41 | 56.02 | 59.83 | 64.03 | 65.96 | 67.08 |
| ReLU (Sensoy et al., 2018) | 22.46 | 27.83 | 31.39 | 33.18 | 35.61 | 37.03 | 38.95 |
| ReLU (Sensoy et al., 2018) + MEH | 22.46 | 29.10 | 32.71 | 34.95 | 37.16 | 38.47 | 39.84 |
| Softmax | 52.53 | 58.26 | 61.13 | 65.05 | 66.44 | 68.41 | 69.55 |
| Softmax + MEH | 52.53 | 59.68 | 64.78 | 67.80 | 68.28 | 70.85 | 71.84 |
| **Softmax + MEH + HUA** (Ours) | **52.53** | **62.02** | **66.84** | **69.82** | **71.31** | **72.86** | **74.08** |

Table 1: mAP (%) of RetinaNet on PASCAL VOC.

**Comparison of computing costs.** In Table 2, we compare computing costs of the proposed method with multi-model based baselines (MC-dropout, *Ensemble*) and *Entropy* baseline. Specifically, the first column indicates the latency time for uncertainty calculation of a single image with a 95% confidence interval and the second column shows the size of all parameters in the models. It can be seen that the proposed method has a similar model size to that of MC-dropout and Ensemble baselines; a slight increase is due to the MEH module. With respect to the latency time, multi-model based methods require a significant computing cost because of multiple forward propagations for MC integration. Specifically, MC-dropout and *Ensemble* take 25 and 3 forward passes, respectively. In comparison, our method requires much less latency time, although it samples 500 ensemble members $Cat(\theta)$. This is because our method requires only single forward propagation and each ensemble member is sampled virtually (i.e., $\theta \sim Dir(\theta|\alpha)$). Note that *Entropy* has the lowest latency, since this method dose not use epistemic uncertainty and there is no need to sample ensemble members.

**Visualization analysis of model evidence $\lambda$.** To clearly understand the role of model evidence $\lambda$, we illustrate the effect of $\lambda$ in Fig. 5. In the $2^{nd}$ column, when model evidence $\lambda$ is not considered, uncertain regions are observed over a wide area inside the object. On the other hand, in the $3^{rd}$ column, MEH estimates high $\lambda$ only around the center of objects. This is because the difficulty of classification increases as the center of the bounding box moves away from the center of objects due to ambiguity. The MEH output, $\lambda$, is used to readjust the prior Dirichlet distribution $Dir(\theta|\alpha)$. As noted in Section 3.2, a high $\lambda$ increases Dirichlet strength and make $D(\theta|\alpha)$ concentrated. It makes similar $\theta \sim Dir(\theta|\alpha)$ be sampled and decreases the epistemic uncertainty according to equation (1).

| Method | RetinaNet | | SSD | |
|---|---|---|---|---|
| | Unc. calculation (img/sec) | Model size (MB) | Unc. calculation | Model size |
| Ensemble | $0.1387 \pm 0.0036$ | 420.882 | $0.0837 \pm 0.0023$ | 300.813 |
| MC-dropout | $1.1058 \pm 0.0048$ | 140.294 | $0.6184 \pm 0.0032$ | 100.271 |
| Entropy | $0.0445 \pm 0.0013$ | 140.294 | $0.0256 \pm 0.0013$ | 100.271 |
| Proposed | $0.0629 \pm 0.0022$ | 149.377 | $0.0310 \pm 0.0010$ | 100.781 |

Table 2: Comparison of computing costs on PASCAL VOC.

For example, in the $4^{th}$ column, it can be seen that the uncertainty of the central part of the object is reduced due to high $\lambda$; conversely, the uncertainty of the edge part of the object is preserved high.

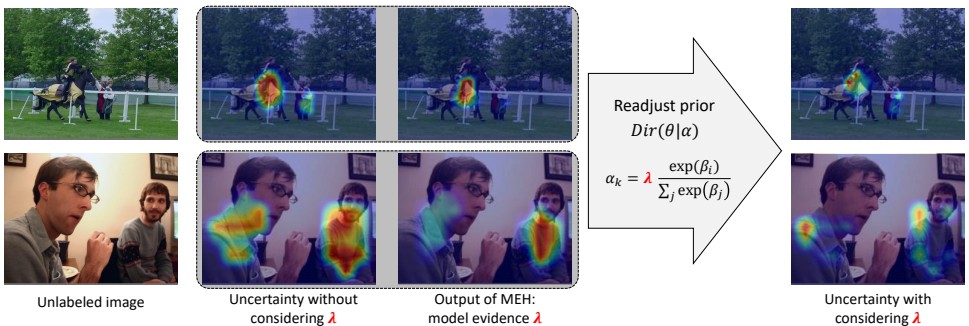

Figure 5: Effect of model evidence $\lambda$ when calculating epistemic uncertainty. Warm color indicates high value. In the area with high model evidence, epistemic uncertainty decreases.

**Examples of easy and hard samples.** In Fig. 6, for intuitive understanding, we compare uncertainties of easy and hard samples with *Entropy Sampling* (Roy et al., 2018), and MI-AOD (Yuan et al., 2021) which represent SoTA. In all methods, easy examples tend to have just one unoccluded object; however, hard examples tend to have numerous objects, which are unclear or heavily occluded. One interesting thing to note is that difference in informativeness scores between the easy samples and the hard samples. In previous methods, the range of scores between images is not wide, since these methods average the uncertainties of all bounding boxes to get a single score. On the other hand, in the proposed method, it can be seen that the informativeness scores of the hard examples are more than 50 times higher than those of the easy examples. This is because the proposed uncertainty aggregation scheme (HUA) adds up the informativeness scores of meaningful bonding boxes with consideration for additional information (e.g., the number of objects, categories in images).

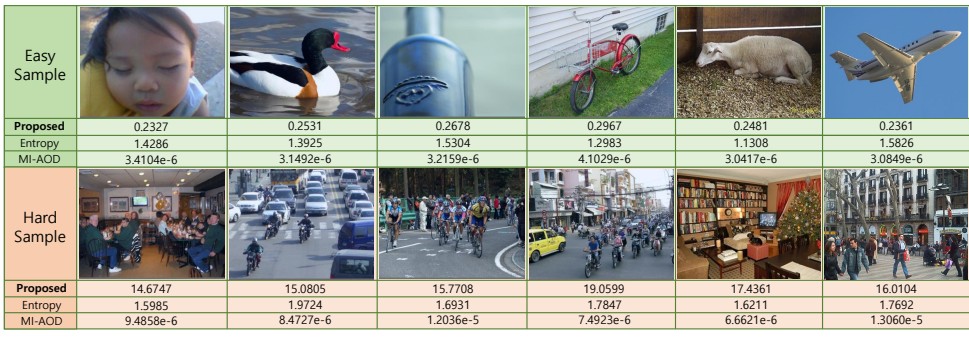

Figure 6: Examples of easy and hard samples. Informativeness scores are reported below images.

## 5 CONCLUSION

We proposed a new active learning strategy for object detection. With the evidential deep learning framework and model evidence head, our scheme effectively estimates the epistemic uncertainty of a bounding box. Also, our hierarchical uncertainty aggregation provides a new guideline for computing informativeness of an image. Our scheme presents an up-and-coming direction for active object detection, where estimating epistemic uncertainty accurately yet quickly is of crucial importance.

## REPRODUCIBILITY STATEMENT

To ensure reproducibility, we provide theoretical proof and implementation details in Appendix. Specifically, our experimental setup including hyperparameter setting is described in Appendix, and our source codes are included in Supplementary Material. The code is available at https://github.com/MoonLab-YH/AOD_MEH_HUA

### 5.1 ACKNOWLEDGEMENT

This work was conducted by Center for Applied Research in Artificial Intelligence (CARAI) grant funded by DAPA and ADD (UD190031RD). Also, this work was supported by Institute of Information & Communications Technology Planning & Evaluation(IITP) funds from MSIT of Korea (No. 2020-0-00626).

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

## A    APPENDIX

## B    ADDITIONAL EXPERIMENTAL SETUP

In this section, we provide more detailed information on our experimental setting. Following previous works, two well-known object detectors are chosen as base models: RetinaNet (Lin et al., 2017) and SSD (Liu et al., 2016). First, RetinaNet adopts ResNet-50 which is pretrained on ImageNet as backbone. ResNet-50 has four stages and output of each stage is passed to the neck structure. The output dimension of each stage is 256, 512, 1024, 2048, respectively. The parameters of the backbone in the first stages are frozen during training. As for the neck structure, Feature Pyramid Network (FPN) with five stages is adopted. FPN takes the output of ResNet as input and returns 256-dimensional features at the every stage. The output of FPN then goes through a classification head and a regression head, independently. The structure of the classification head and the regression head are the same as that of vanilla RetinaNet (Lin et al., 2017) which consists of three convolutional modules and one last convolutional layer for classification and regression, respectively. Following (Lin et al., 2017; Yuan et al., 2021), we utilize a bounding box generator whose octave base scale is 4; scales per octave is 3; ratios of bounding boxes are (0.5, 1.0, 2.0). The base sizes of boxes are (8, 16, 32, 64, 128) from the lowest scale to the highest scale. The coordinates of bounding boxes and ground truth are encoded to the delta and considered as the target value of the regression head. Specifically, given central coordinates $(x, y)$, width $(w)$ and height $(h)$ are given for the bounding box $(px, py, pw, ph)$ and the ground truth $(gx, gy, gw, gh)$, values of delta is computed as: $dx = \frac{gx-px}{pw}, dy = \frac{gy-py}{ph}, dw = log(\frac{gw}{pw}), dh = log(\frac{gh}{ph})$. Regression heads are supervised to predict the encoded value $(gx, gy, gw, gh)$ and these values are decoded to the coordinates of the uppermost left corner and the lowermost right corner to make the final prediction. Regarding the loss, as stated in the main paper, focal loss (Lin et al., 2017) with marginal likelihood is adopted. Hyperparameters for focal loss $(\gamma, \alpha)$ are set to 2.0 and 0.25, respectively. As to assigning positive/negative boxes, bounding boxes whose IoU scores with the ground truth are higher than 0.5 are assigned to the positive box; boxes with iou scores greater than 0.4 and less than 0.5 are assigned as negative boxes and all remaining boxes are ignored. As for evaluation, 1000 bounding boxes with the highest confidence are used for Non-Maximum Suppression (NMS). The IoU threshold for NMS is set to 0.5 and the number of boxes after NMS is at most 100. Secondly, SSD adopts VGG-16 which is pretrained on ImageNet as backbone. Following (Liu et al., 2016), the backbone of SSD has extra layers to generate multi-scale feature maps. After these layers, six levels of features are created with dimensions of (512, 1024, 512, 256, 256, 256) and these features are taken as inputs to the classification head and the regression head. The same encoding/decoding scheme as RetinaNet was used. Different from RetinaNet, hard negative mining is applied to assign positive/negative boxes; specifically, three times more negative boxes than positive boxes are assigned. As for evaluation, the same NMS scheme is adopted as RetinaNet except the number of bounding boxes after NMS is set to 200. For both models, the SGD optimizer is used with learning rate of 0.001; a momentum of 0.9 and a weight decay of 0.0001.

## C    ALGORITHM DESCRIPTION

The detailed procedure of the proposed active object detection method is given in Algorithm 1.

## D    PROOF OF MARGINAL LIKELIHOOD LOSS

We consider the Dirichlet-Categorical Bayesian framework where the low-order predictive distribution parameters $\theta$ are sampled from the high-order prior distribution (i.e., $\theta \sim Dir(\theta|\alpha)$ and $\theta$ parameterizes categorical likelihood $Cat(\theta)$.) To train our evidential model, we maximize the marginal likelihood which is given as below:

$$p(y_i|\alpha) = \int p(y_i|\theta)p(\theta|\alpha)d\theta = \int \prod_{j=1}^{K} \theta_j^{\mathbb{1}\{y_j=i\}} \frac{\Gamma(\sum_{j=1}^{K} \alpha_j)}{\prod_{j=1}^{K} \Gamma(\alpha_j)} \prod_{j=1}^{K} \theta_j^{\alpha_j-1} d\theta \qquad (7)$$

---

**Algorithm 1** Proposed Active Learning Method

---

**Input:** Initialized model $\mathbf{\Phi}^0 = [\mathbf{\Phi}_D^0, \mathbf{\Phi}_{MEH}^0]$ where $\mathbf{\Phi}_D$ denotes all model parameters excluding MEH, labeled set $\mathbf{L}^0$, unlabeled set $\mathbf{U}^0$

**Output:** Model $\mathbf{\Phi}^T = [\mathbf{\Phi}_D^T, \mathbf{\Phi}_{MEH}^T]$ after $T$ active learning cycles

**for** each active learning cycle $t = 0, 1, ..., T-1$ **do**

**Step 1: Train the model (including MEH module) using $\mathbf{L}^t$**

1: **for** each labeled data $x_l \in L^t$ **do**
2:      Predict class confidence $\beta = \{\beta_1, \beta_2, ..., \beta_K\}$ and model evidence $\lambda$
3:      Compute concentration parameter $\alpha = \{\alpha_k = \frac{\lambda exp(\beta_k)}{\sum_i exp(\beta_i)}|k = 1, 2, ..., K\}$
4:      Compute predictive probability $p(y = k|\alpha) = \frac{\alpha_k}{\sum_i \alpha_i}$
5:      Compute classification loss $L_{cls}(y, p(y = k|\alpha))$
6:      Update object detector $\mathbf{\Phi}_D^t$ using $L_{cls}$
7:      Compute MEH loss $L_{MEH}(\lambda, L_{cls})$
8:      Update MEH $\mathbf{\Phi}_{MEH}^t$ using $L_{MEH}$ with $\mathbf{\Phi}_D^t$ fixed
9: **end for**

**Step 2: Compute informativeness of unlabeled samples via EDL and HUA**

1: **for** each unlabeled data $x_u \in U^t$ **do**
2:      **for** each bounding box **do**
3:          Predict class confidence $\beta = \{\beta_1, \beta_2, ..., \beta_K\}$ and model evidence $\lambda$
4:          Compute concentration parameter $\alpha = \{\alpha_k = \frac{\lambda exp(\beta_k)}{\sum_i exp(\beta_i)}|k = 1, 2, ..., K\}$
5:          Sample categorical distributions as $\theta \sim Dir(\theta|\alpha)$
6:          Compute epistemic uncertainty $\mathcal{I}[y, \theta] = \mathcal{H}\big[\mathbb{E}_{p(\theta|\alpha)}[p(y|\theta)]\big] - \mathbb{E}_{p(\theta|\alpha)}\big[\mathcal{H}[p(y|\theta)]\big]$
7:      **end for**
8:      Aggregate $\mathcal{I}[y, \theta]$ of all bounding boxes via HUA to get informativeness $\mathcal{I}(x_u)$
9: **end for**

**Step 3: Select informative data and update $\mathbf{L}^t$, $\mathbf{U}^t$**

1: Select the most informative data $\mathbf{I}^t \subset \mathbf{U}^t$ based on $\mathbf{I}(\mathbf{U}^t) = \{\mathbf{I}(x_u)|x_u \in \mathbf{U}^t\}$
2: Human oracles make labels of $\mathbf{I}^t$
3: $\mathbf{L}^{t+1} = \mathbf{L}^t \cup \mathbf{I}^t$             // Update labeled set
4: $\mathbf{U}^{t+1} = \mathbf{U}^t n \mathbf{I}^t$           // Update unlabeled set

**end for**

---

which is equivalent to

$$\frac{\Gamma(\sum_{j=1}^{K} \alpha_j)}{\prod_{j=1}^{K} \Gamma(\alpha_j)} \int \prod_{j=1}^{K} \theta_j^{\mathbb{1}\{y_j=i\}} \prod_{j=1}^{K} \theta_j^{\alpha_j - 1} d\theta = \frac{\Gamma(\sum_{j=1}^{K} \alpha_j)}{\prod_{j=1}^{K} \Gamma(\alpha_j)} \int \prod_{j=1}^{K} \theta_j^{\alpha_j + \mathbb{1}\{y_j=i\} - 1} d\theta \qquad (8)$$

The above integral can be simplified using a useful trick. Given $Dir(\theta|\alpha)$, integrating probability density function over $\theta$ results in one:

$$\int \frac{\Gamma(\sum_{j=1}^{K} \alpha_j)}{\prod_{j=1}^{K} \Gamma(\alpha_j)} \prod_{j=1}^{K} \theta_j^{\alpha_j - 1} d\theta = \frac{\Gamma(\sum_{j=1}^{K} \alpha_j)}{\prod_{j=1}^{K} \Gamma(\alpha_j)} \int \prod_{j=1}^{K} \theta_j^{\alpha_j - 1} d\theta = 1 \qquad (9)$$

Dividing the above equation by a constant part, which is not involved in the integration, gives the following result:

$$\int \prod_{j=1}^{K} \theta_j^{\alpha_j - 1} d\theta = \frac{\prod_{j=1}^{K} \Gamma(\alpha_j)}{\Gamma(\sum_{j=1}^{K} \alpha_j)} \qquad (10)$$

Now we can regard equation (8) as the integration over "shifted parameter" (i.e., $\alpha_j + \mathbb{1}\{y_j = i\}$). This shift of parameter changes the equation (10) to:

$$\int \prod_{j=1}^{K} \theta_j^{\alpha_j + \mathbb{1}\{y_j=i\} - 1} d\theta = \frac{\prod_{j=1}^{K} \Gamma(\alpha_j + \mathbb{1}\{y_j = i\})}{\Gamma(\sum_{j=1}^{K}(\alpha_j + \mathbb{1}\{y_j = i\}))} \qquad (11)$$

Considering the case when $y_j = i$ separately, the above equation (11) can be written as:

$$\frac{\prod_{j=1}^{K} \Gamma(\alpha_j + \mathbb{1}\{y_j = i\})}{\Gamma(\sum_{j=1}^{K}(\alpha_j + \mathbb{1}\{y_j = i\}))} = \frac{\prod_{j \neq i}^{K} \Gamma(\alpha_j)\Gamma(\alpha_i + 1)}{\Gamma(\sum_{j=1}^{K} \alpha_j) + 1)} \qquad (12)$$

Now we can use the fact that $\Gamma(x+1) = x\Gamma(x)$ to simplify the above equation as:

$$\int \prod_{j=1}^{K} \theta_j^{\alpha_j + \mathbb{1}\{y_j=i\}-1} d\theta = \frac{\prod_{j \neq i}^{K} \Gamma(\alpha_j)\Gamma(\alpha_i+1)}{\Gamma(\sum_{j=1}^{K}(\alpha_j)+1)} = \frac{\prod_{j=1}^{K}\Gamma(\alpha_j)}{\Gamma(\sum_{j=1}^{K}(\alpha_j))} \cdot \frac{\alpha_i}{\sum_{j=1}^{K}(\alpha_j)} \quad (13)$$

By substituting the result of the above equation in the integral of equation (8), we obtain:

$$p(y_i|\alpha) = \int p(y_i|\theta)p(\theta|\alpha)d\theta = \frac{\Gamma(\sum_{j=1}^{K}\alpha_j)}{\prod_{j=1}^{K}\Gamma(\alpha_j)} \cdot \frac{\prod_{j=1}^{K}\Gamma(\alpha_j)}{\Gamma(\sum_{j=1}^{K}(\alpha_j))} \cdot \frac{\alpha_i}{\sum_{j=1}^{K}(\alpha_j)} = \frac{\alpha_i}{\sum_{j=1}^{K}(\alpha_j)} \quad (14)$$

## E  ANALYSIS OF THRESHOLD VALUES

To understand HUA better, we investigate the effect of each threshold value: $\gamma_{score}, \gamma_{IoU}$. We vary the thresholds and report the results in Table 3. Note that aggregate functions (Sum, Max, Sum) are applied to information levels (Object, Scale, Class) in all tries for fair comparison. It can be seen that higher $\gamma_{IoU}$ increases the performance. This is because excessively low thresholds cause too many unnecessary bounding boxes to be taken into account, thus deteriorating the quality of estimated informativeness. Otherwise, a too high or too low $\gamma_{score}$ impairs the quality of filtering and degrades the final performance.

| Thresholds | | Ratio (%) of Labeled Samples | | | | | | |
|---|---|---|---|---|---|---|---|---|
| $\gamma_{score}$ | $\gamma_{IoU}$ | 5.0 | 7.5 | 10.0 | 12.5 | 15.0 | 17.5 | 20.0 |
| 0.2 | 0.7 | 52.53 | 60.40 | 66.35 | 69.19 | 70.82 | 72.11 | 73.32 |
| 0.2 | 0.8 | 52.53 | 60.62 | 66.44 | 69.22 | 70.59 | 72.52 | 73.63 |
| 0.2 | 0.9 | 52.53 | 60.95 | 66.68 | 69.71 | 70.90 | 72.69 | 73.71 |
| 0.3 | 0.7 | 52.53 | 61.16 | 66.09 | 69.29 | 70.98 | 72.57 | 73.80 |
| 0.3 | 0.8 | 52.53 | 61.54 | 66.62 | 69.57 | 71.28 | 72.70 | 74.02 |
| **0.3** | **0.9** | **52.53** | **62.02** | **66.84** | **69.82** | **71.31** | **72.86** | **74.08** |
| 0.4 | 0.7 | 52.53 | 59.43 | 65.84 | 67.92 | 70.07 | 71.90 | 73.15 |
| 0.4 | 0.8 | 52.53 | 61.30 | 65.35 | 67.82 | 70.66 | 71.69 | 73.35 |
| 0.4 | 0.9 | 52.53 | 60.75 | 65.68 | 68.66 | 71.09 | 71.81 | 73.78 |

Table 3: Performance (mAP%) of RetinaNet on PASCAL VOC.

## F  ANALYSIS OF HUA

Fig. 7 visualizes how HUA works in detail. Note that aggregation functions (Sum, Max, Sum) are chosen for (Object, Scale, Class). First, it can be seen that unclear or occluded objects have high uncertainty. For example in the left image, occluded boats in the left side show higher uncertainty than the one in the right side. It can also be seen that uncertainties of numerous objects within the left image are summed and contribute to the higher informativeness score. Note that since the aggregation function for "Class" is "Sum", bounding box which is classified into multiple classes simultaneously is supposed to have high uncertainty. For example in the right image, the head of horse in the right image is classified into the horse and the cow at the same time; bounding boxes which predict either category are separated into independent groups, and uncertainty for each category group is computed independently and summed together.

## G  COMPARISON WITH OTHER LOSS TYPE FOR MEH

For the training of MEH, the mean Square Error (**MSE**) loss is adopted for regression of the target loss. In this section, we further investigate performances of other typical losses which are well-known for the regression problem. Table 4 compares the mAP50 score on PASCAL VOC with RetinaNet as the base model. The method **MAE** indicates the mean absolute error; given prediction $\hat{y}$ and ground truth $y$, $Loss(y, \hat{y}) = |y - \hat{y}|$. The **MSLE** method refers to the mean squared logarithmic error, (i.e., $Loss(y, \hat{y}) = (log(y+1) - log(\hat{y}+1))^2$). Experimental results show that general losses for regression are all effective, yet MSE loss shows the best result.

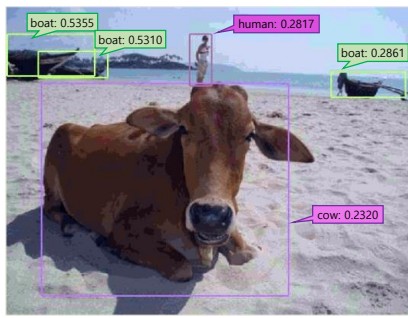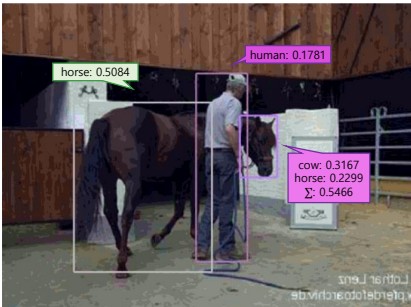

Informativeness : $\sum_j (Unc)_j$ = 1.8663      Informativeness : $\sum_j (Unc)_j$ = 1.2331

Figure 7: Epistemic uncertainty of bounding boxes and eventual informativeness score of images.

| Loss Type | Ratio (%) of Labeled Samples | | | | | | |
|---|---|---|---|---|---|---|---|
| | 5.0 | 7.5 | 10.0 | 12.5 | 15.0 | 17.5 | 20.0 |
| MAE | 52.53 | 59.55 | 65.65 | 68.78 | 70.54 | 71.80 | 73.50 |
| MSLE | 52.53 | 61.50 | 66.76 | 69.43 | 70.76 | 72.20 | 72.95 |
| **MSE** | **52.53** | **62.02** | **66.84** | **69.82** | **71.31** | **72.86** | **74.08** |

Table 4: Performance (mAP%) of RetinaNet on PASCAL VOC.

## H  ANALYSIS ABOUT THE SELECTION RATIO OF FILTERED SAMPLES

In this section, we compare performance of the proposed method varying the selection ratio of filtered samples. Note that the bounding boxes whose confidence value are lower than $\gamma_{score}$ are filtered out to ignore predictions for background. This scheme naturally makes the informativeness score of any image to zero, if all bounding boxes in the image are unconfident. We stress that these images with zero informativeness are important since the object detector is unable to find any object due to its lack of knowledge. In Table 5, it can be seen that taking some portion of these images of zero informativeness into account helps to improve the performance. This is because training with filtered images provides the model to experience difficult objects from unfamiliar categories. However, including too much filtered images into the labeled set of the next stage rather impair the performance. This is because some objcets with low confidence are too small/large or heavily occluded and interfere with training of the detector.

| Selection Ratio | Ratio (%) of Labeled Samples | | | | | | |
|---|---|---|---|---|---|---|---|
| | 5.0 | 7.5 | 10.0 | 12.5 | 15.0 | 17.5 | 20.0 |
| 5% | 52.53 | 61.07 | 66.15 | 68.09 | 70.26 | 72.24 | 72.80 |
| 10% | 52.53 | 62.25 | 66.23 | 69.55 | 70.96 | 71.57 | 73.12 |
| **15%** | **52.53** | **62.02** | **66.84** | **69.82** | **71.31** | **72.86** | **74.08** |
| 20% | 52.53 | 62.28 | 66.86 | 69.42 | 71.27 | 72.36 | 73.56 |
| 30% | 52.53 | 60.54 | 65.73 | 69.02 | 71.13 | 72.38 | 73.50 |
| 50% | 52.53 | 59.40 | 65.04 | 68.79 | 70.01 | 70.26 | 72.46 |

Table 5: Performance (mAP%) of RetinaNet on PASCAL VOC.

## I   COMPARISON OF AGGREGATION FUNCTION

An obvious question in HUA is: which combination of aggregation functions yields the best performance? In Table 6, we compare three types of functions: Avg, Max, Sum. In all tries, $\gamma_{score}, \gamma_{IoU}$ is set to 0.3, 0.9, respectively. We try various combinations applying different aggregation functions in each level of information. It can be seen (Sum, Max, Sum) gives the best performance exhibiting a large margin over naive aggregation schemes of (Choi et al., 2021; Yuan et al., 2021). Besides, *sum* operation is observed to perform the best for "Object", since it reflects the number of objects inside an image. Likewise, *sum* operation shows the best result for "Class". It confirms the importance of considering uncertainties from ambiguous classes together. Otherwise, *max* operation prevails for "Scale". Since predictions can appear at various scales depending on the shape of the object regardless of the amount of information, it seems that the quality of uncertainty is rather impaired if all scales are considered.

| Function | Ratio (%) of Labeled Samples | | | | | | |
|---|---|---|---|---|---|---|---|
| (Obj, Scale, Class) | 5.0 | 7.5 | 10.0 | 12.5 | 15.0 | 17.5 | 20.0 |
| (Avg, Avg, Avg) (Yuan et al., 2021) | 52.53 | 60.04 | 63.89 | 67.30 | 69.25 | 70.03 | 71.12 |
| (Avg, Avg, Sum) | 52.53 | 60.71 | 62.64 | 66.01 | 68.58 | 69.80 | 69.89 |
| (Avg, Sum, Sum) | 52.53 | 59.74 | 62.17 | 64.79 | 67.50 | 68.92 | 70.82 |
| (Avg, Max, Sum) | 52.53 | 59.78 | 63.50 | 66.52 | 67.85 | 69.38 | 70.03 |
| (Max, Avg, Sum) | 52.53 | 61.09 | 65.60 | 68.98 | 70.25 | 71.70 | 72.21 |
| (Max, Max, Max) (Choi et al., 2021) | 52.53 | 60.29 | 64.52 | 68.13 | 69.06 | 71.53 | 72.60 |
| (Sum, Sum, Sum) | 52.53 | 61.35 | 66.09 | 69.06 | 70.17 | 71.94 | 73.09 |
| (Sum, Max, Avg) | 52.53 | 61.12 | 65.19 | 67.89 | 70.45 | 72.28 | 73.10 |
| (**Sum**, **Max**, **Sum**) | **52.53** | **62.02** | **66.84** | **69.82** | **71.31** | **72.86** | **74.08** |

Table 6: Performance (mAP %) of RetinaNet on PASCAL VOC.

## J   DETECTION OF NOISY LABELS

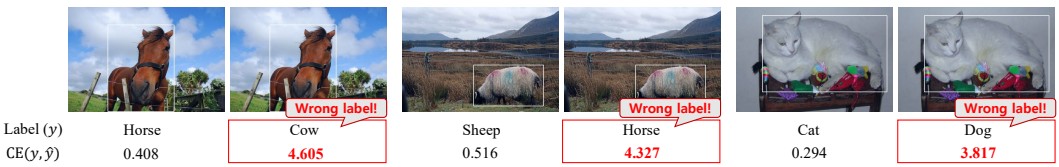

Figure 8: Noisy label detection via cross-entropy.

In active learning work, generally, it is assumed that a perfect oracle generates 100% accurate labels to unlabeled samples, so that the main focus of existing active learning works (including our work) becomes effectively seleting the informative samples under the assumption with no label noise. In this section, we deal with an interesting question: how should we handle the label noise issue when human annotations are not 100% accurate? Orthogonal to the existing active learning methods, a simple way is to utilize a new module that identifies the noisy labels and re-label them or exclude them during training. As illustrated in works of (Bernhardt et al., 2022; Younesian et al., 2021), cross-entropy is commonly adopted to detect the mislabeled samples in noise-robust learning literatures. To demonstrate the capability our model for detecting noisy labels, we present simple toy examples in Figure 8. To reproduce the noisy label, we change labels ($y$) of some bounding boxes to other similiar categories (e.g., horse → cow), and compute the cross entropy using the probability distribution ($\hat{y}$) of Equation 3 for comparison. As can be seen in Figure 8, bounding boxes with wrong labels show much higher cross entropy, ensuring opportunities to cope with noisy labels.

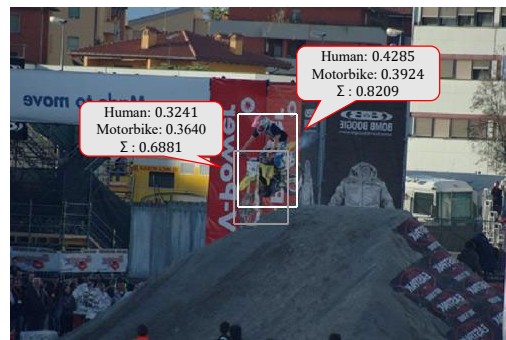 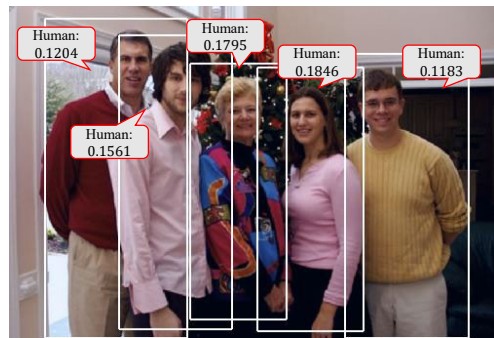

Informativeness : $\Sigma_i (Epistemic\ Unc.) = 1.5090$    Informativeness : $\Sigma_i (Epistemic\ Unc.) = 0.7589$

Figure 9: Epistemic uncertainty of bounding boxes and resulting informativeness scores of images.

## K   ANALYSIS ON HUA WITH RESPECT TO THE NUMBER OF OBJECTS

As can be seen in Figure 6, HUA calculates image informativeness considering various information inside the image (number of objects, difficulty of each object), which makes the uncertainty scores of hard examples significantly higher than those of easy examples. However, this characteristic of HUA raises the following question: Does HUA favor only complex images with many objects? Wouldn't it be possible to replace HUA by simply selecting images in which many objects appear? To answer these question, we investigated the relationship between the number of objects in the image and the uncertainty score, and found that a large number of objects does not always lead to a high uncertainty score, as shown in Figure 9. This is because the resulting uncertainty of an image depends not only on the number of objects but also on the **uncertainty value of each object**. For example, in the image on the left of Figure 9, there exists two objects, but the total uncertainty is higher than that of the right image, because the bounding boxes corresponding to the objects are classified as people and motorbikes at the same time. On the other hand, since the objects in the right image are easy objects that are clearly classified only as people, the uncertainty of all five objects is not too large.

## L   ANALYSIS ON THE ORDER OF HUA

In this section, we investigate the effect of the aggregation order in HUA to provide additional insight. Under the same conditions as the experiment conducted in the main manuscript, we only changed the order of scale and class, i.e., the aggregation hierarchy in HUA is changed from "object (sum) ← scale (max) ← class (sum)" to "object (sum) ← class (sum) ← scale (max)". As shown in the table below, the performance is degraded compared to the proposed scheme when we switch the order.

| Method | Ratio (%) of Labeled Samples | | | | | | |
|---|---|---|---|---|---|---|---|
| | 5.0 | 7.5 | 10.0 | 12.5 | 15.0 | 17.5 | 20.0 |
| Reversed Order | 52.53 | 62.42 | 66.42 | 68.69 | 70.58 | 72.06 | 72.98 |
| Proposed Order | 52.53 | 62.02 | 66.84 | 69.82 | 71.31 | 72.86 | 74.08 |

Table 7: mAP (%) of RetinaNet on PASCAL VOC.

We interpret this result as follows: 1) Since most objects do not span multiple scales (they mostly span $1 \sim 2$ scales), there is little difference even if the order of class and scale have changed. 2) If the order is changed, uncertainty values from multiple scales may be mixed into a single informativeness score of one object. Rather, it seems more appropriate to focus on a single informative scale to compute the score of the image. Overall, both our intuition and experimental results indicate that the original order is more powerful and makes more sense.

# M ANALYSIS ON GENERALITY OF HUA

To demonstrate the generality of the proposed HUA, we conducted additional experiments. We combined traditional methods (Roy et al., 2018; Yuan et al., 2021) with HUA and reported the performance gain in Table 8.

| Method | Ratio (%) of Labeled Samples | | | | | | |
|---|---|---|---|---|---|---|---|
| | 5.0 | 7.5 | 10.0 | 12.5 | 15.0 | 17.5 | 20.0 |
| Entropy (Roy et al., 2018) | 29.13 | 49.41 | 56.02 | 59.83 | 64.03 | 65.96 | 67.08 |
| Entropy (Roy et al., 2018) + HUA | 29.13 | 50.24 | 56.91 | 60.5 | 65.17 | 66.49 | 68.11 |
| MIAOD (Yuan et al., 2021) | 47.18 | 58.11 | 63.72 | 67.32 | 69.39 | 70.83 | 71.91 |
| MIAOD (Yuan et al., 2021) + HUA | 47.18 | 59.07 | 64.61 | 68.76 | 70.04 | 71.95 | 72.63 |

Table 8: mAP (%) of RetinaNet on PASCAL VOC.

As shown in the table, HUA generally improves the performance in both cases. This is because HUA helps to better understand the context within images by considering the information hierarchy (e.g., class, object, scale), than just taking the maximum value of the uncertainties of all bounding boxes as done in (Roy et al., 2018; Yuan et al., 2021). This result confirms the intuition that existing works can be enhanced by adopting HUA for uncertainty aggregation.

# N APPLICATION TO A TWO-STAGE DETECTOR

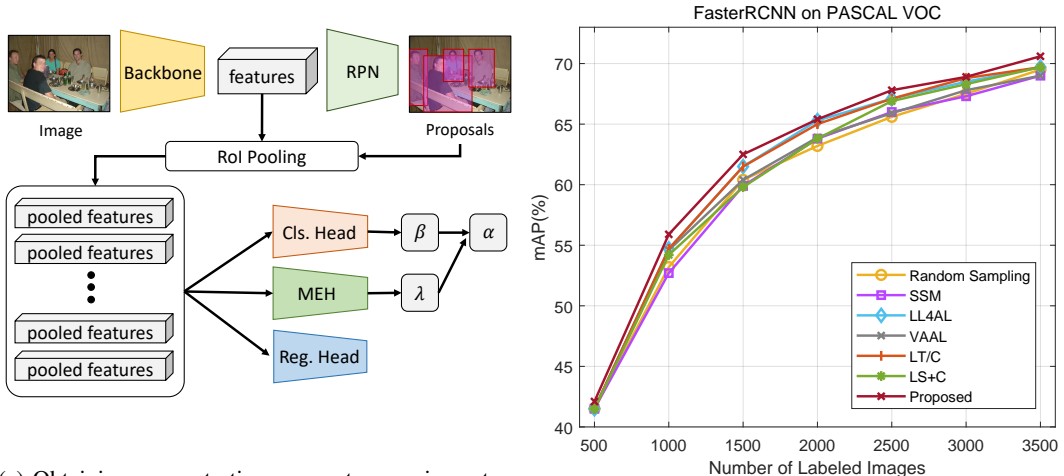

(a) Obtaining concentration parameter $\alpha$ using a two-stage detector

(b) Performance of FasterRCNN on PASCAL VOC

Figure 10: Proposed EDL-based uncertainty computation of a bounding box using a **two-stage detector**. (a) First, for an unlabeled image, the backbone network and RPN produce features and proposals, respectively. Taken ROI pooled features as input, classification head generates class confidences $\beta = \{\beta_i\}_{i=1}^K$ while our model evidence head (MEH) produces model evidence $\lambda$. $\beta$ and $\lambda$ are used to compute a parameter set $\alpha$ of Dirichlet distribution $Dir(\theta|\alpha)$. (b) Performance (mAP %) of FasterRCNN on PASCAL VOC

Although we have verified our algorithms (MEH, HUA) only for single-stage detectors (RetinaNet, SSD) in the main paper, these algorithms can be easily extended to two-stage detectors as well. Fig. 10a shows the high-level descriptions of our approach under the EDL framework in a two-stage detector. For each proposal, the ROI pooled features are taken as input by the classification head and MEH. Then class confidence $\beta = \{\beta_k\}_{k=1}^K$ and model evidence $\lambda$ are generated, and concentration parameter $\alpha$ of the prior Dirichlet distribution $Dir(\theta|\alpha)$ can be computed via equation (4). Given $\theta \sim Dir(\theta|\alpha)$, epistemic uncertainty of every proposal is computed via equation (1). Afterwards, epistemic uncertainty of all proposals are aggregated considering information hierarchy (e.g., predicted class, scale, number of detected object) through HUA to get total uncertainty of the image. The only difference from the single-stage detector is that it computes uncertainty only for

proposed bounding boxes by RPN rather than all bounding boxes. Fig.10b shows the performance in PASCAL VOC 2012 when the proposed methods are applied to FasterRCNN (Ren et al., 2015). It can be seen that the proposed method performs better than various baselines (SSM(Wang et al., 2018), LS+C and LT/C(Kao et al., 2018), VAAL(Sinha et al., 2019), LL4AL(Yoo & Kweon, 2019)), which confirms that our scheme can give general performance gain even in the two-stage detector.

