# OpenReview forum: "Active Learning for Object Detection with Evidential Deep Learning and Hierarchical Uncertainty Aggregation"
_ICLR.cc/2023/Conference — ICLR 2023 poster_

### Official Review · Reviewer_P2kB · 2022-10-23

**Confidence:** 3
**Correctness:** 4
**Technical Novelty And Significance:** 4
**Empirical Novelty And Significance:** 4
**Recommendation:** 6

**Clarity, Quality, Novelty And Reproducibility:**

* Clarity:

  * Clearly written and well-structured paper overall.
  * Minor observations:
    * Section 3.3, subsection on re-aligning bounding boxes: "each bounding box contains much more information:
    <b>object</b>, scale and category". As output by the detector, bounding boxes at each scale do not come with an
    assigned object identifier, nor are ground truth object bounding box annotations available during active learning
    to perform an assignment. Rather, if my understanding is correct, detections are first clustered based on an IoU
    criterion, and each cluster becomes an object hypothesis, with several detections assigned to it. I think this
    should be clarified.
    * The middle quantity
    $\frac{\Gamma(\alpha_j + 1)\Gamma(\sum_j \alpha_j)}{\Gamma(\alpha_j)\Gamma(\sum_j \alpha_j + 1)}$ in equation (3)
    seems to be wrong (it does not depend on $i$, while the left hand side $p(y_i \mid \alpha)$ and
    right hand side both do).
    * Typos and grammar:
      * Section 3.2: "applying EDL to objection to describe" => "applying EDL to object detection and describe"
      * Section 4.1: "at the every cycle" => "at every cycle"
      * Section 4.3: "focal coss" => "focal loss", "as din in" => "as done in"

* Novelty:

  * Up to my knowledge, this is the first work to apply EDL to object detection. It also brings several technical
  innovations to the method, as discussed in the paper summary. A formal motivation of the MEH loss and a discussion
  of modelling epistemic uncertainty for bounding box localization would add further novelty.

* Quality:

  * Relevant baselines and benchmarks are considered, comparisons are fair, and ablation studies are provided. Overall,
  the scientific quality of this work seems adequate.

* Reproducibility:

 * The authors mention the relevant details needed to replicate their work.
 * The authors provide full source code in the supplementary material.

**Strength And Weaknesses:**

* Strengths:

  * High potential impact, given the relatively little work on Active Learning for object detection and the high
  practical need for such methods in the industry (ubiquity of object detection in practice, high annotation costs)
  * Low inference time makes this method quite attractive compared to other epistemic uncertainty evaluation approaches
  * Extensive evaluation, providing a fair comparison against relevant alternative methods, while also providing
  ablation studies and runtime comparisons.
  * The paper is well-structured and easy to follow.

* Weaknesses:

  * While the authors provide an intuition for their loss function for MEH (decrease model evidence when the bounding
  box classification error is large and vice-versa), no probabilistic interpretation is given to this loss within the
  EDL framework. The lack of a formal derivation makes it look somewhat empirical and disconnected to the rest of the
  EDL framework.
  * The MEH is attached to the classification head. However, epistemic uncertainty is not modelled for the regression
  head (bounding box localization), and the method relies on the presence of multiple detections as a proxy for
  localization uncertainty. While such improvements may be out of scope, I think the possibility of modelling epistemic
  uncertainty jointly in classification/localization space should be discussed.

**Summary Of The Paper:**

The authors tackle the problem of model uncertainty estimation as the acquisition function in Active Learning for object
detection. To this end, they first adapt Evidential Deep Learning (EDL) to predict the epistemic uncertainty estimate
for each bounding box separately. They modify EDL in three ways: (a) replace the last layer's ReLU by softmax,
(b) introduce a separate Model Evidence Head (MEH) that predicts model evidence, effectively re-scaling the
concentration parameter and (c) change the loss function to encourage the model evidence to be inversely proportional to
the classification loss. Once uncertainty is estimated for each bounding box, the authors introduce a three-level
hierarchical scheme to aggregate uncertainty across bounding boxes by iteratively merging them based on their predicted
classes, scales and spatial overlap (objects).

Experiments carried out on Pascal VOC and MS-COCO show superiority to competing state-of-the-art approaches in active
learning (final mAP when reaching a fixed proportion of labelled samples in a pool-based active learning simulation).
Ablation studies are also provided showing the relative improvement of each design decision and computational
complexity.

**Summary Of The Review:**

Overall, I think this is good quality work, with sufficient technical contribution and good empirical support.
The technical contribution could further be strengthened by providing a formal motivation for the MEH loss and a
discussion of how one could model  epistemic uncertainty for the bounding box localization predictor head. Minor
improvements to clarity/corrections should be considered.

---

> ### Author Response · Authors · 2022-11-16
> **Response to Reviewer P2kB**
>
> First of all, we appreciate the reviewer for the time and efforts, and proving comments that could further strengthen our paper. We also appreciate the reviewer for acknowledging our work to be of high quality with sufficient technical contributions. Below, we provide responses to the reviewer's comments.
>
> &nbsp;
>
>
> ### **Probabilistic interpretations in MEH loss:**
>
> We agree with the reviewer that our MEH loss function itself does not directly contain some probabilistic components, since the proposed MEH loss function only focuses on producing the re-scaling parameter $\lambda$. Instead, we would like to position our MEH loss function as follows: The proposed loss function in equation (6) makes EDL more applicable to the “object detection” task based on some clear intuitions. Our solution handles the issues of previous regularization loss in equation (5) that enforces the non-target elements to be one regardless of the error, suppresses the overall model evidence, and makes training unstable. Moreover, the parameter $\lambda$ produced by (6) later re-scales the concentration parameter of the Dirichlet distribution for computing the epistemic uncertainty, which we believe is an important component for the EDL framework.
>
> &nbsp;
>
>
> ### **Modeling epistemic uncertainty with both classification/localization:**
>
>  Thank you for the interesting suggestion. As pointed out by the reviewer, we also feel that measuring  epistemic uncertainty from localization space by applying EDL to Reg. Head is a good direction. To this end, we applied EDL to Reg. Head based on [1]; we predicted the parameters of the Normal Inverse-Gamma distribution, which was the conjugate prior of the Gaussian distribution, but the performance was rather reduced. We observed that the regularization loss in [1] failed to minimize the evidence, so that Reg. Head always produced an overconfident prediction. Specifically, we trained a RetinaNet whose regression head was modified by [1] using the entire MS-COCO dataset, and found that the mAP performance decreased from 36.2% to 34.4% compared to when using the original RetinaNet. Hence, it seems that some sophisticated modifications are required for applying the existing EDL to Reg. Head, just as we improved the idea of [2] for Cls. Head.
>
> [1] Amini, Alexander, et al. "Deep evidential regression." Advances in Neural Information Processing Systems 33 (2020): 14927-14937.
>
> [2] Sensoy, Murat, Lance Kaplan, and Melih Kandemir. "Evidential deep learning to quantify classification uncertainty." Advances in neural information processing systems 31 (2018).
>
>
>
> &nbsp;
>
>
> ###  **Comment on Realigning bounding boxes in Section 3.3:**
> We appreciate your reasonable comment. We changed the expression to “…information: **predicted object**, scale and category…” to emphasize the fact that we exploit the pseudo label (object hypothesis) for HUA.
>
>
> &nbsp;
>
> ###  **Other minor comments:**
>  All reviewer's comments are correct, and thank you for pointing these out. We have corrected the typos in Equation (3), Section 3.2, Section 4.1, and Section 4.3.
>
>
>
>
>
> Overall, we agree with the reviewer's comments and made  efforts to discuss them in our response.
> We hope to know if you are satisfied with our response. We would also like to know if the reviewer has any further questions/concerns on our paper.

---

### Official Review · Reviewer_tNAG · 2022-10-24

**Confidence:** 3
**Correctness:** 3
**Technical Novelty And Significance:** 2
**Empirical Novelty And Significance:** 2
**Recommendation:** 6

**Clarity, Quality, Novelty And Reproducibility:**

The work has its originality. However, the experiments are not comprehensive enough.

**Strength And Weaknesses:**

+ The paper is well-motivated and well-written. Overall the idea is very clear. I can easily understand what the authors want to do.
+ The experiments support the claims. In particular, the proposed method achieves better performance than the baselines.

- The novelty of introducing EDL into active learning seems very straightforward. EDL is able to find some out-of-distribution samples during the labeling process, whether the unlabeled samples are out-of-distribution? This has not been explained clearly by the authors.

- The authors have changed the concentration parameters from greater than 1 to greater than 0, this would affect the Dirichlet distribution. The authors should conduct experiments to show the differences, or at least some toy examples. Moreover, I cannot tell the significance of employing $\lambda$. How do the differences between these two heat maps affect the final labeling? Will the differences impact significantly?

- The authors only employ single-stage detectors to conduct the experiments. Whether the proposed method still works for two-stage detectors?

- As this work focuses on active learning, in each round there would be some randomness. Therefore, the authors should report the variations of each experiment. In other words, by using different labeled samples at the beginning what the variations would be?

- Last but not least, the authors use a certain ratio / percentage of the dataset samples. As different datasets would have different samples and this would make the contributions unclear as to whether the benefits come from the EDL active learning or more labeled samples at the beginning.

- The baselines seems out of date. MI-AOD is published in CVPR 2021, which is the newest one in the comparisons.
Sequential Voting with Relational Box Fields for Active Object Detection, CVPR2022. Note that the github of this work is available.

- The equations in the appendix is not new, which are very similar to EDL paper. Therefore, I do not think adding those equations make this paper valuable as those are not orignal deduction.


**Summary Of The Paper:**

This paper employs evidential deep learning to assign uncertainty to unlabeled samples. Then, based on the assigned uncertainty, active learning is leveraged to ask humans to label more samples from the dataset. The authors have modified the original EDL. Specifically, the concentration parameters $\alpha$ >1 have been changed to $\alpha$>0.  Doing so would make Dirichlet distribution concentrate on sparse samples. The authors also introduced hierarchy uncertainty aggregation to decide the informativeness of an image.

**Summary Of The Review:**

This work is well presented. My two concerns are about the novelty and the experiemnts.
1. The novelty is not clear as I cannot really tell why the difference would make such performance improvements in Fig. 5.
2. The experiments seem not strong: missing two-stage detector, the experiments do not provide variations, the state-of-the-art methods are missing.

---

> ### Author Response · Authors · 2022-11-17
> **Response to Reviewer tNAG (1/2)**
>
> We appreciate the reviewer for helpful comments that are also very clear.  We have worked them into the revised version of the paper. Our responses are given below.
>
> &nbsp;
> ### **Motivation behind adopting EDL:**
> In accordance with the reviewer’s comment, we clarified EDL’s ability to find OOD samples in the introduction of the edited manuscript. We would also like to stress that although EDL has been applied to active learning for classification tasks, existing works have not focused on developing EDL for active learning for object detection, which presents significant challenges.  Our contribution in this paper is to make EDL highly compatible with active learning for object detection, as described in our manuscript and our answers to the reviewer's questions below.
>
> &nbsp;
> ### **Why we have proposed to change the concentration parameter $\alpha$:**
> The reason we changed the concentration parameters ($\alpha$) from greater than 1 to greater than 0 was to enable confident prediction. According to Equations 2\&3 in the manuscript, predictive probability for classification is defined using the ratio of $\alpha$ as $p(y=i|\alpha)=\frac{\alpha_i}{\sum{\alpha_k}}$. As proposed in [1], if $\alpha$ is forced to be greater than 1 as $\alpha_k=ReLU(\beta_k)+1$, the denominator of $p(y|\alpha)$ becomes too large and consequently the prediction becomes excessively underconfident. Although it does not significantly affect the performance (accuracy) in image classification, it considerably reduces the performance (mAP) in object detection. This discussion can be found in page 4 of our original paper.
>
> The effect of changing ReLU to softmax for the concentration parameter can be found in Table 1 of our original manuscript: with 20\% of labeled samples, the performance increases from 38.95\% to 69.55\% by replacing ReLU to softmax.
> Overall, our contribution in this work is to adopt Softmax to handle the issue that arises from the naive application of EDL [1] to object detection. We are ready to answer any other questions the reviewer might have regarding the effects of the proposed components.
>
> &nbsp;
> ### **Significance of employing $\lambda$ in Fig. 5 and performance improvement:**
> The model evidence  $\lambda$  we introduced is significant because it transforms the Dirichlet distribution (Dir($\theta|\alpha$)) by reflecting the expected difficulty and, as a result, it helps to obtains accurate epistemic uncertainty of bounding boxes. For example, let's say MEH generates a high value of $\lambda$. Although $\lambda$ does not change the prediction $p(y=i|\alpha)=\alpha_i/\sum{\alpha_k}$, it sharpens Dir($\theta|\alpha)$, and consequently reduces the epistemic uncertainty in equation 1. This intention can also be confirmed in Fig. 1(b) of the original manuscript.
>
> Fig. 5 shows how our intentions are observed in real images. Before going into further, note that the color at one coordinate in the heatmap reflects the average value of epistemic uncertainty/model evidence of bounding boxes whose centers correspond to that coordinate. Compared to the 2nd column, it can be seen that the uncertainty of the center parts of objects decreases when $\lambda$ is considered (4th column). This is because MEH generates high $\lambda$ by predicting low difficulty in the central part of the object. Consequently, introducing $\lambda$ encourages to focus on bounding boxes which are really difficult for the current model (the edge part rather than the central part).
>
> The effect of employing $\lambda$ can be also seen from Table 1 of our manuscript: by comparing "softmax" and "softmax + MEH", it can be seen that the model performance is increased from 69.55\% to 71.84\% by introducing model evidence $\lambda$.
>
> &nbsp;
> ### **Applicability to two-stage detectors (Updated in Section N of Appendix):**
> We appreciate the reviewer for the question. Although we have verified our algorithms (MEH, HUA) only for single-stage detectors (RetinaNet, SSD) in the main paper, these algorithms can be easily extended to existing two-stage detectors as well.
> Whereas MEH/HUA were applied to all bounding boxes within the image in single-stage detectors, we can apply MEH/HUA only to boxes in RoI (region of interest) proposed by RPN (Region Proposal Network) in two-stage detectors.  We illustrated a detailed model overview in Figure 10(a) of Appendix.
>
> According to the reviewer’s question, we combine our methods with Faster-RCNN and reported its performance in Figure 10(b) of Appendix. We utilize FasterRCNN as a base model and compared the performance in PASCAL VOC with various baselines (SSM, LS+C, LT/C, VAAL, LL4AL).  As can be seen in Figure 10(b), even though we couldn't completely fine-tune the hyperparameters due to the time constraints, our proposed method performs better than the baselines
> confirming that our scheme is still effective in two-stage detectors.

---

> > ### Author Response · Authors · 2022-11-17
> > **Response to Reviewer tNAG (2/2)**
> >
> > ### **Randomness/variations in our experiments (Updated in Figure 3 of the manuscript):**
> > As correctly pointed out by the reviewer, active learning generally measures the average of multiple experiments with fixed seeds to cope with the randomness.
> > Based on the setup of [1], [2], [3], we did collect experimental results using 5 fixed seed sets, and report the average values. In each seed, the identical initial labeled set \& model initialization is used for all baselines for a fair comparison. When we consider another seed, the initial labeled set \& model initialization are different from the previous seed to capture the variations and randomness. These discussions was provided in "Baselines" in page 7 of our manuscript. According to the reviewer's comment, we additionally illustrate the 95\% confidence interval of the experimental results in Figure 3 of the manuscript so that the variations of each experiment are more clearly revealed.
> >
> > &nbsp;
> > ### **Ratio/percentage of the dataset samples:**
> > Thanks for the interesting question.  We would first like to note that for all experiments,  we followed the standard setups of  [1], [2] and set the ratios accordingly. We also utilized the same ratio value for all baselines in each experiment, for a fair comparison.
> >
> > Nevertheless, according to the review, we conducted additional experiments applying the same ratio/percentage conditions to PASCAL VOC as MS-COCO. Specifically, just as in MS-COCO, 2345 (2\% of 117,267) labeled samples of PASCAL VOC were used in the initial cycle and 2345 samples were added into the labeled set every cycle. As can be seen in the table below, the proposed algorithm shows a consistent trend outperforming other baselines.
> >
> > ##### &nbsp;&nbsp;&nbsp;&nbsp;&nbsp;&nbsp;&nbsp;&nbsp;&nbsp;&nbsp;&nbsp;&nbsp;&nbsp;&nbsp;&nbsp;&nbsp;&nbsp;&nbsp; mAP (\%) of RetinaNet on PASCAL VOC
> > | Number of Labeled Samples  | 2,345 | 4,690 | 7,035 | 9,380 | 11,725 |
> > |:----------:|:-------------:|:------:|:------:|:------:|:------:|
> > | Random | 64.02 | 70.40 | 72.22 | 74.53 | 76.28 |
> > | MI-AOD  | 67.58 | 73.66 | 75.18 | 77.13 | 78.17 |
> > | **Proposed** | **64.02** | **76.14** | **78.30** | **79.15** | **80.52** |
> >
> > ##### &nbsp;&nbsp;&nbsp;&nbsp;&nbsp;&nbsp;&nbsp;&nbsp;&nbsp;&nbsp;&nbsp;&nbsp;&nbsp;&nbsp;&nbsp;&nbsp;&nbsp;&nbsp;&nbsp;&nbsp;&nbsp;&nbsp;&nbsp; mAP (\%) of SSD on PASCAL VOC
> > | Number of Labeled Samples  | 2,345 | 4,690 | 7,035 | 9,380 | 11,725 |
> > |:----------:|:-------------:|:------:|:------:|:------:|:------:|
> > | Random | 60.21 | 66.51 | 68.45 | 69.13 | 71.05 |
> > | MI-AOD | 62.43 | 68.81 | 72.34 | 73.55 | 74.58 |
> > | **Proposed** | **60.21** | **68.84** | **73.52** | **74.76** | **76.43** |
> >
> > &nbsp;
> > ### **Additional up-to-date baseline (Updated in Figure 3 of the edited manuscript.):**
> > We appreciate the reviewer for recommending a new baseline. We updated Figure 3 of our revised manuscript. We first tried to choose the paper recommended by the reviewer “Sequential Voting with Relational Box Fields for Active Object Detection”  as a new baseline, but we realized that this work does not focus on active learning for object detection, but focuses on finding the active object, the object that is being manipulated by the human hand.
> >
> > Instead, we report the performance of “Consistency-based Active Learning method for object Detection (CALD)” presented in CVPR 2022 with PASCAL VOC and MS-COCO in Figure 3. It can be seen that our method is still competitive or performs better than these baselines, confirming the advantage of the proposed ideas. By adding this baseline, we currently believe that we have considered most of the recent works that could be applied to "active learning for object detection", but we are also open to add any other baselines suggested by the reviewers that are relevant to our work.
> >
> > &nbsp;
> > ### **Equations in Appendix:**
> > We admit the equations (equations 2, 3 in the manuscript) we proved are not new. However, our intention behind the proof in Appendix was not to claim the novelty, but to help readers more easily understand the mathematical foundation of equations, because the authors of [3] did not provide proofs for the equation. We will make this point clear.
> >
> > [3] Sensoy, Murat, Lance Kaplan, and Melih Kandemir. "Evidential deep learning to quantify classification uncertainty." Advances in neural information processing systems 31 (2018).
> >
> > &nbsp;
> >
> > Again, we appreciate the reviewer for the time and effort. Your concerns are clear and to the point, and we feel we have successfully clarified all the issues you raised. Specifically, we described why the effect of $\lambda$ in Fig. 5 improves the performance, showed applicability to two-stage detectors, added a baseline published in CVPR 2022, and provided variations for each scheme. In case there are remaining questions/concerns, we hope to be able to have an opportunity to further answer them.

---

### Official Review · Reviewer_4FYw · 2022-10-24

**Confidence:** 4
**Correctness:** 3
**Technical Novelty And Significance:** 3
**Empirical Novelty And Significance:** 3
**Recommendation:** 6

**Clarity, Quality, Novelty And Reproducibility:**

The work of this paper is of high quality, and the authors point out the shortcomings of traditional methods and gives better solutions. Experiments show that the proposed method brings a high performance improvement, which demonstrates the effectiveness of the method. The paper is well presented; however, there is still some ambiguity as mentioned in the weaknesses.


**Strength And Weaknesses:**

Strength:

- EDL is applied to predict the epistemic uncertainty of samples.
- The authors propose a HUA method to aggregate object-level uncertainty into image-level uncertainty.
- Mathematical proofs make this method more rigorous and convincing.
The proposed method shows good performance.

Weaknesses:

- There are some points unclear on the uncertainty score. Is the epistemic uncertainty (Eq. 1) directly used as the uncertainty score of the box? If so, does this bring better results? Does this mean that object-level uncertainty does not need to account for aleatoric uncertainty.
- Algorithm 1 seems to be just an overview of an active learning framework. The core part of active learning seems not present. The authors may need to describe Step 2 in detail: how to calculate the box uncertainty and how to aggregate the box uncertainty, which can make the algorithm clearer.
- There seems no experiments to tell how about combining traditional methods and HUA.
- The results of "Examples of easy and hard samples" are interesting. The uncertainty scores of complex images are significantly higher than those of simple images. However, does this lead to selecting only complex images and ignoring simple images each time? The images that are ultimately selected tend to be those that contain more objects. It seems that the active selection strategy is unnecessary. We may not need the epistemic uncertainty, but only need to roughly estimate which images contain more objects. Are there any further experiments to have a discussion?
- Do the results obained using only MEH (without HUA) have a clear advantage over others? It would be better to have experiments on this point. Otherwise, it is difficult to prove the validity of epistemic uncertainty.


**Summary Of The Paper:**

This paper combines evidential deep learning (EDL) and active learning in the object detection task and achieve good results. The authors propose a new module model evidence head (MEH) and a new loss function to ensure that EDL can be effectively applied to object detection. For active learning, a hierarchical uncertainty aggregation (HUA) method is proposed to obtain the informativeness of an image. Experiments show that the proposed method brings a very good performance improvement.


**Summary Of The Review:**

The paper cleverly combines EDL and active learning in the object detection task. It would be better to address the weaknesses.

---

> ### Author Response · Authors · 2022-11-16
> **Response to Reviewer 4FYw (1/2)**
>
> We thank the reviewer for the efforts as well as the valuable comments. We also appreciate the reviewer for acknowledging our work as a high quality paper. Regarding the concerns/questions raised, we believe we successfully addressed every single one, as described below.
>
> &nbsp;
> ### **Regarding the uncertainty score in equation 1:**
>  Yes, epistemic uncertainty in equation 1 is directly used for the uncertainty score of a bounding box.  Note that in previous works that adopt non-Bayesian frameworks, ``aleatoric uncertainty = total uncertainty'' holds and thus epistemic uncertainty is not considered. In contrast, since we consider a Bayesian framework, it can be seen from equation 1 that epistemic uncertainty reflects the values of both aleatoric uncertainty and the total uncertainty. In other words, aleatoric uncertainty is already taken into account in our epistemic uncertainty value described in equation 1.
>
> &nbsp;
> ### **Detailed description of Algorithm 1 (Updated in Section C of Appendix):**
> We appreciate this comment. We now improved our pseudo code algorithm in Algorithm 1. We specifically described the steps for computing the bounding box uncertainty and hierarchical uncertainty aggregation process.
>
> &nbsp;
> ### **Combining HUA with other existing methods (Updated in Section M of Appendix):**
> Thank you for pointing out this. To demonstrate the generality of the proposed HUA, we conducted additional experiments. According to the comment, we combined traditional methods [1], [2] with HUA and reported the performance gain in the table below.
>
> ##### &nbsp;&nbsp;&nbsp;&nbsp;&nbsp;&nbsp;&nbsp;&nbsp;&nbsp;&nbsp;&nbsp;&nbsp;&nbsp;&nbsp;&nbsp;&nbsp;&nbsp;&nbsp;&nbsp;&nbsp;&nbsp;&nbsp;&nbsp;&nbsp;&nbsp;&nbsp;&nbsp;&nbsp;&nbsp;&nbsp;&nbsp;&nbsp;&nbsp;&nbsp;&nbsp;&nbsp;&nbsp;&nbsp;&nbsp; mAP (\%) of RetinaNet on PASCAL VOC
> | Ratio of Labeled Samples  |    5.0\% |  7.5\% | 10.0\% | 12.5\% | 15.0\% | 17.5\% | 20.0\% |
> |:----------:|:-------------:|:------:|:------:|:------:|:------:|:------:|:-----:|
> | Entropy [1]|  29.13 | 49.41 | 56.02 | 59.83 | 64.03 | 65.96 | 67.08 |
> | Entropy [1]+HUA | 29.13 | 50.24 | 56..91 | 60.50 | 65.17 |  66.49 | 68.11 |
> | MIAOD [2]| 47.18 | 58.11 | 63.72 | 67.32 | 69.39 | 70.83 | 71.91 |
> | MIAOD [2]+HUA| 47.18 | 59.07 | 64.61 | 68.76 | 70.04 | 71.95 | 72.63 |
>
> As shown in the table, HUA generally improves the performance in both cases. This is because HUA helps to better understand the context within images by considering the information hierarchy (e.g., class, object, scale), than just taking the maximum value of the uncertainties of all bounding boxes as done in [1], [2]. This result confirms the intuition that existing works can be enhanced by adopting HUA for uncertainty aggregation. We added this analysis in the Section M of the Appendix.
>
> [1] Roy, Soumya, Asim Unmesh, and Vinay P. Namboodiri. "Deep active learning for object detection." BMVC. 2018.
>
> [2] Yuan, Tianning, et al. "Multiple instance active learning for object detection." Proceedings of the IEEE/CVF Conference on Computer Vision and Pattern Recognition. 2021.

---

> > ### Author Response · Authors · 2022-11-16
> > **Response to Reviewer 4FYw (2/2)**
> >
> > ### **Considering only the number of objects for labeling (Updated in Section K in Appendix):**
> > This is an interesting question, and we appreciate it. We must stress that HUA doesn’t always favor images with many objects. The total uncertainty of an image computed by HUA depends not only on the number of objects but also on the ``uncertainty value of each object''. In Section K of the Appendix, we have added these discussions along with some examples. In Figure 9, it can be seen that the left image with one human riding a motorbike has higher uncertainty than the image on the right with five humans.
> >
> > In accordance with the reviewer’s comment, we also performed additional experiments for the scheme that only cares about the number of objects within the image for labeling:  it prioritizes the images which contain more objects for active selections. As can be seen in the table below, the proposed HUA outperforms the baseline confirming that HUA can better estimate informativeness scores using various information other than the number of objects (e.g., epistemic uncertainty, size of object, detected category, etc.).
> >
> > ##### &nbsp;&nbsp;&nbsp;&nbsp;&nbsp;&nbsp;&nbsp;&nbsp;&nbsp;&nbsp;&nbsp;&nbsp;&nbsp;&nbsp;&nbsp;&nbsp;&nbsp;&nbsp;&nbsp;&nbsp;&nbsp;&nbsp;&nbsp;&nbsp;&nbsp;&nbsp;&nbsp;&nbsp;&nbsp;&nbsp;&nbsp;&nbsp;&nbsp;&nbsp;&nbsp;&nbsp;&nbsp;&nbsp;&nbsp; mAP (\%) of RetinaNet on PASCAL VOC
> > | Ratio of Labeled Samples  |    5.0\% |  7.5\% | 10.0\% | 12.5\% | 15.0\% | 17.5\% | 20.0\% |
> > |:----------:|:-------------:|:------:|:------:|:------:|:------:|:------:|:-----:|
> > | Only considering the number of object |  **52.53** | 61.12 | 64.04 | 67.34 | 69.29 | 70.96 | 72.14 |
> > | **Proposed HUA** | **52.53** | **62.02** | **66.84** | **69.82** | **71.31** | **72.86** | **74.08** |
> >
> > &nbsp;
> > ### **Performance of only MEH (without HUA):**
> > The performance of ``Softmax + MEH'' in Table 1 of our main manuscript corresponds to the scheme with only MEH (without HUA). Comparing this results with the results in Fig. 3(a) and Fig. 4(a), it can be seen that MEH already performs better than most of the baselines including CDAL, coreset, random sampling, entropy sampling, MC dropout, model ensemble, confirming the advantage of using epistemic uncertainty.
> >
> > &nbsp;
> > Again, thank you for your valuable comments. Please do let us know if you have any remaining questions/comments.

---

### Official Review · Reviewer_HUy5 · 2022-10-25

**Confidence:** 4
**Correctness:** 3
**Technical Novelty And Significance:** 3
**Empirical Novelty And Significance:** 3
**Recommendation:** 6

**Clarity, Quality, Novelty And Reproducibility:**

The quality of the paper is good: it studies the fundamental task, object detection in computer vision, and has achieved the new state-of-the-art results.

The clarity of the paper is good: introduction of the method, experiments and claims are clear enough to read and understand.


Source code is provided in the supplementary material for reproducibility.

**Strength And Weaknesses:**

Strengths:

1. This paper aims to label a set of samples based on the uncertainty by adopting Evidential Deep Learning (EDL) to effectively compute epistemic uncertainty, and proposes the Model Evidence Head (MEH) that makes EDL highly compatible with object detection.
2. The proposed Hierarchical Uncertainty Aggregation (HUA) makes use of attributes in bounding boxes, such as nearest object and box size, to capture the context within the image and improves the quality of the expected informativeness of images, instead of simply depending on the maximum/mean of all bounding boxes.

Weaknesses:

1. The motivation of MEH module is not clear, eg., why applying softmax function instead of ReLU operation can produce sufficiently confident prediction? Would replacing ReLU with sigmoid function can help MEH module?
2. The lack of the denotation of symbol \beta_{c} in equation 4.
3. How to compute the large error in equation 6?
4. Can authors provide the performance of variants of HUA where the order of `scale and class’ is switched.
5. Why the concentration parameter cannot be measured by Reg.Head but for Cls. Head?




**Summary Of The Paper:**

This work fist considers epistemic uncertainty for capturing the usefulness of the sample by model evidence head (MEH) which is a kind of evidential deep learning (EDL), and then proposes hierarchical uncertainty aggregation (HUA) for obtaining the informativeness of an image. Experimental results show the effectiveness of proposed method.

**Summary Of The Review:**

I would recommend a marginally above acceptance. The experimental results and reported performances are the state-of-the-arts and may have a good impact to the field of classical objection detection, but i would like to know the influences in the DETR-like models.

---

> ### Author Response · Authors · 2022-11-16
> **Response to Reviewer HUy5 (1/2)**
>
>  We appreciate the reviewer for the thoughtful comments. We also appreciate the reviewer for acknowledging the contribution of our work in object detection. Below, we reply to the comments raised by the reviewer.
>
> &nbsp;
>
> ### **Motivation of MEH:**
>
> We appreciate the reviewer for the comment on the motivation of replacing ReLU with softmax in MEH, and also suggesting the sigmoid function. Although ReLU in [1] performs well on image classification tasks with simple datasets, it noticeably decreases the mAP score in object detection where the number of classes is large and confident prediction is important. For example, in the case of 80-way classification, $\beta_k$ should be at least $7820$ to achieve $\hat{p_k}=0.99$ even when $\beta_{i}=0$ for all $i\neq k$, according to  $\alpha_k'=\textrm{ReLU}(\beta_k)+1$ and  $p(y_i|\alpha) =   \frac{\alpha_i}{\sum_j\alpha_j}$.  These discussions are provided in page 4 of our manuscript, and this is the reason why ReLU does not perform well even when combined with our MEH module.
>
> Similarly, it is difficult to obtain sufficiently confident predictions with sigmoid ($\alpha_k = sigmoid(\beta_k)$) for the following reasons: 1) Since the sigmoid function is bounded to 1, it is difficult for a certain class to have a particularly higher $\alpha_k$ than other classes. 2) In sigmoid, the sum of class confidence for each class ($\sum \alpha_k$) is not 1. For example, there possibly be $\alpha_1 = 0.8, \alpha_2 = 0.7$. These two limitations of sigmoid reduce the magnitude of $\alpha_k$ of each class, and prevent obtaining sufficiently confident predictions via equation 3. To verify this, we replaced ReLU with sigmoid and compare the performance under the same conditions, and the results are as follows.
>
> ##### &nbsp;&nbsp;&nbsp;&nbsp;&nbsp;&nbsp;&nbsp;&nbsp;&nbsp;&nbsp;&nbsp;&nbsp;&nbsp;&nbsp;&nbsp;&nbsp;&nbsp;&nbsp;&nbsp;&nbsp;&nbsp;&nbsp;&nbsp;&nbsp;&nbsp;&nbsp;&nbsp;&nbsp;&nbsp;&nbsp;&nbsp;&nbsp;&nbsp;&nbsp;&nbsp;&nbsp;&nbsp;&nbsp;&nbsp; mAP (\%) of RetinaNet on PASCAL VOC
> | Ratio of Labeled Samples  |    5.0\% |  7.5\% | 10.0\% | 12.5\% | 15.0\% | 17.5\% | 20.0\% |
> |:----------:|:-------------:|:------:|:------:|:------:|:------:|:------:|:-----:|
> | Random |  29.13 | 50.35 | 58.45 | 61.27 | 62.31 | 64.67 | 66.72 |
> | Sigmoid | 52.51 |  59.92 | 65.44 | 67.72  | 71.01 |  71.96 | 72.82 |
> | **Proposed** | **52.53** | **62.02** | **66.84** | **69.82** | **71.31** | **72.86** | **74.08** |
>
> It can be seen that EDL with sigmoid shows higher performance than vanilla Retinanet, but lower performance than the proposed method (softmax); unlike ReLU and sigmoid, the softmax function adopted in this paper is more suitable to EDL for object detection because it is not upper-bounded and has a good property that the sum of predicted values ​​for all classes is 1.  Our contribution in this paper is to adopt softmax for a more confident prediction and then design MEH that re-scales the concentration parameter, to handle the issue that arises from the naïve application of EDL to object detection.
>
> [1] Sensoy, Murat, Lance Kaplan, and Melih Kandemir. "Evidential deep learning to quantify classification uncertainty." Advances in neural information processing systems 31 (2018).
>
> &nbsp;
>
> ###  **$\beta_c$ in equation 4:**
>
> $\beta_c$ is the class confidence for class $c$, which is produced from  the classification head. We made this clearer in the revised manuscript.
>
> &nbsp;
>
> ### **Equation 6:**
>
>  In equation 6,  $l_{s,i}$ is   a classification loss for bounding box $i$ at scale $s$ (e.g., the focal loss).  Our MEH loss in equation 6   prevents overconfidence and helps the model to predict uncertainty better; when $l_{s,i}$ is large, the MEH loss in equation 6 guides to decrease $\lambda_{s,i}$ while MEH is guided to increase $\lambda_{s,i}$ for a small $l_{s,i}$.  For a clearer presentation, we changed our original expression "when large error $l_{s,i}$ is detected" to  "when the classification loss $l_{s,i}$ is large".

---

> > ### Author Response · Authors · 2022-11-16
> > **Response to Reviewer HUy5 (2/2)**
> >
> > ### **Switching the order of ''scale and class'' in HUA (Updated in Section L of Appendix):**
> >
> > This is an interesting point. We performed additional experiments on this. Under the same conditions as the experiment conducted in the main manuscript, we only changed the order of scale and class, i.e., the aggregation hierarchy in HUA is changed from "object (sum) $\leftarrow$ scale (max) $\leftarrow$ class (sum)" to "object (sum) $\leftarrow$ class (sum) $\leftarrow$ scale (max)".
> > As shown in the table below, the performance is degraded compared to the proposed scheme when we switch the order.
> >
> > ##### &nbsp;&nbsp;&nbsp;&nbsp;&nbsp;&nbsp;&nbsp;&nbsp;&nbsp;&nbsp;&nbsp;&nbsp;&nbsp;&nbsp;&nbsp;&nbsp;&nbsp;&nbsp;&nbsp;&nbsp;&nbsp;&nbsp;&nbsp;&nbsp;&nbsp;&nbsp;&nbsp;&nbsp;&nbsp;&nbsp;&nbsp;&nbsp;&nbsp;&nbsp;&nbsp;&nbsp;&nbsp;&nbsp;&nbsp; mAP (\%) of RetinaNet on PASCAL VOC
> > | Ratio of Labeled Samples  |    5.0\% |  7.5\% | 10.0\% | 12.5\% | 15.0\% | 17.5\% | 20.0\% |
> > |:----------:|:-------------:|:------:|:------:|:------:|:------:|:------:|:-----:|
> > | Reversed Order| 52.53| 62.42 | 66.42 | 68.69 | 70.58 | 72.06 | 72.98 |
> > | Proposed Order| 52.53| 62.02 | 66.84 | 69.82 | 71.31 | 72.86 | 74.08 |
> >
> > We interpret this result as follows:
> > 1) Since most objects do not span to multiple scales (they mostly span to 1 $\sim$ 2 scales), there is little difference even if the order of class and scale have changed.
> > 2) If the order is changed, uncertainty values ​​from multiple scales may be mixed into a single informativeness score of one object. Rather, it seems more appropriate to focus on a single informative scale to compute the score of the image. Overall, both intuitively and experimentally,  the original order is more powerful and makes more sense. Based on the reviewer's comment, we added these interesting discussions in the Appendix.
> >
> >
> > &nbsp;
> >
> > ### **Measuring concentration parameter by Reg. Head:**
> >
> > In our current paper, we measured the concentration parameter from Cls. Head instead of Reg. Head, because the concentration parameter $\alpha$  is a parameter of the Dirichlet distribution that is the prior of the categorical distribution. We believed that the information from Reg. Head, which was about continuous values, was not adequate to predict $\alpha$. But aside from that, the reviewer's idea of measuring uncertainty using Reg. Head seems to make sense. To this end, we applied EDL to Reg. Head based on [1]. We predicted the parameters of the Normal Inverse-Gamma distribution, which was the conjugate prior of the Gaussian distribution, but the performance was rather reduced. We observed that the regularization loss in [1] failed to minimize the evidence, so that Reg. Head always produced an overconfident prediction. Specifically, we trained a RetinaNet whose regression head was modified by [1] using the entire MS-COCO dataset, and found that the mAP performance decreased from 36.2% to 34.4% compared to when using the original RetinaNet. Hence, it seems that some sophisticated modifications are required to apply the existing EDL to Reg. Head, just as we improved the idea of [2] for Cls. Head.
> >
> > [1] Amini, Alexander, et al. "Deep evidential regression." Advances in Neural Information Processing Systems 33 (2020): 14927-14937.
> >
> > [2] Sensoy, Murat, Lance Kaplan, and Melih Kandemir. "Evidential deep learning to quantify classification uncertainty." Advances in neural information processing systems 31 (2018).
> >
> > We again thank the reviewer for providing very helpful comments. If there are any more clarifications we could provide, we would be grateful if you could let us know.

---

### Official Review · Reviewer_boP4 · 2022-10-25

**Confidence:** 5
**Correctness:** 4
**Technical Novelty And Significance:** 3
**Empirical Novelty And Significance:** 3
**Recommendation:** 6

**Clarity, Quality, Novelty And Reproducibility:**

- Clarity: the paper is well-written and the idea is very clear.
- Quality: the paper is in a good quality but the experimental validation is still insufficient.
- Novelty: intermediate novelty.
- Reproducibility: the source code is provided in the supplementary.

**Strength And Weaknesses:**

## Strengths
- Overall the paper is well written and easy to follow.
- The idea of MEH and HUA is interesting.
- The experimental results are solid and look promising.

## Weaknesses
- Why both ReLU and ReLU + MEH perform worse than the random baseline in Table 1?
- Is the proposed method able to handle the label noise? Usually we cannot guarantee that the human annotations are 100% accurate and therefore a solid active learning method should also be able to handle the label noise to some extent.
- The baselines of active learning are not enough. The authors should compare with the latest methods not limited to:

[1] Yoo, Donggeun, and In So Kweon. "Learning loss for active learning." In Proceedings of the IEEE/CVF conference on computer vision and pattern recognition, pp. 93-102. 2019.

[2] Kim, K., Park, D., Kim, K.I. and Chun, S.Y., 2021. Task-aware variational adversarial active learning. In Proceedings of the IEEE/CVF Conference on Computer Vision and Pattern Recognition (pp. 8166-8175).

[3] Yu, Weiping, Sijie Zhu, Taojiannan Yang, and Chen Chen. "Consistency-based active learning for object detection." In Proceedings of the IEEE/CVF Conference on Computer Vision and Pattern Recognition, pp. 3951-3960. 2022.


**Summary Of The Paper:**

The paper proposed a model evidence head (MEH) and a hierarchical uncertainty aggregation (HUA) for active selecting informative images under the evidential deep learning (EDL) framework. The experiments conducted on PASCAL VOC and MS-COCO seem to validate the efficacy of the proposed method.


**Summary Of The Review:**

I initially rated the paper as "5: marginally below the acceptance threshold". After carefully reading the authors' response, I am glad to see that they have addressed most of my concerns and therefore I would like to upgrade my rating to "6: marginally above the acceptance threshold".

---

> ### Author Response · Authors · 2022-11-16
> **Response to Reviewer boP4 (1/2)**
>
> We would like to thank the reviewer for the efforts and constructive suggestions, which have greatly helped us to improve the paper.  Below, we provide answers to the reviewer's comments.
>
> &nbsp;
>
> ### **Why both ReLU and ReLU + MEH achieve low performance:**
>
> The reason why ReLU-based schemes  perform poorly in Table 1 is that, in a nutshell,
>   ReLU is not suitable for evidential deep learning (EDL) for object detection, resulting in even
>   worse performance than the random baseline.  Although ReLU performs well on classification tasks with simple datasets [1], [2], [3], it noticeably degrades the mAP score in “object detection” where the number of classes is large and confident prediction is important. For example, in the case of 80-way classification, $\beta_k$ should be at least $7820$ to achieve $\hat{p_k}=0.99$ even when $\beta_{i}=0$ for all $i\neq k$, according to  $\alpha_k'=\textrm{ReLU}(\beta_k)+1$ and  $p(y_i|\alpha) =   \frac{\alpha_i}{\sum_j\alpha_j}$.
>  These discussions are provided in page 4 of our manuscript, and this is the reason why ReLU does not perform well even when combined with our MEH module. Our approach is to adopt softmax for a more confident prediction and then design MEH that re-scales the concentration parameter, to handle the issue that arises from the naïve application of EDL to object detection.
>
>  [1] Sensoy, Murat, Lance Kaplan, and Melih Kandemir. "Evidential deep learning to quantify classification uncertainty." NeurIPS 2018.
>
>  [2] Hemmer, Patrick, Niklas Kühl, and Jakob Schöffer. "Deal: deep evidential active learning for image classification." Deep Learning Applications, Volume 3. Springer, Singapore, 2022. 171-192.
>
>  [3] Zhao, Xujiang, et al. "Uncertainty aware semi-supervised learning on graph data." NeurIPS 2020.
>
> &nbsp;
>
> ### **How to handle label noise (Updated in Section J of Appendix):**
>
> We appreciate the reviewer for the interesting/important question. In the active learning literature, it is generally assumed that a perfect oracle generates 100\% accurate labels, so that the main focus of existing active learning works (including our work) was to effectively select the informative samples under the assumption of no label noise.
>
> However, since the human oracle cannot always be accurate, the reviewer’s question is relevant for the development of practical active learning solutions.  Orthogonal to the existing active learning methods, a simple way is to utilize a new module that identifies the noisy labels and re-label them or exclude them during training. As illustrated in [4], [5], cross-entropy is commonly adopted to detect the mislabeled samples in the noise-robust learning literature. To demonstrate the capability of our model in detecting noisy labels, in Section J of Appendix, we present some simple toy examples. To reproduce the noisy label, we change labels of some bounding boxes
> to other similar categories (e.g., horse$\rightarrow$cow), and compute the cross entropy using the predictive
> distribution for comparison. As can be seen in the figure, bounding boxes with
> wrong labels show much higher cross-entropy, giving opportunities to cope with noisy labels.
>
>  [4] Bernhardt, M., Castro, D. C., Tanno, R., Schwaighofer, A., Tezcan, K. C., Monteiro, M., ... \& Oktay, O. (2022). Active label cleaning for improved dataset quality under resource constraints. Nature communications, 13(1), 1-11.
>
>  [5] Younesian, T., Zhao, Z., Ghiassi, A., Birke, R., \& Chen, L. Y. (2021, November). QActor: Active Learning on Noisy Labels. In Asian Conference on Machine Learning (pp. 548-563). PMLR.

---

> > ### Author Response · Authors · 2022-11-16
> > **Response to Reviewer boP4 (2/2)**
> >
> > ### **Additional up-to-date baselines (Updated in Figure 3 of the edited manuscript):**
> >
> > We appreciate for pointing out recent baselines. First, we would like to note that the performance of [6] (LL4AL) was already reported in Figure 3 of the manuscript. Next, we tried to measure the performance of [7] (TA-VAAL) for a new baseline, but [7] was actually a paper about image classification \& semantic segmentation, which cannot be simply applied to object detection. Instead, in accordance with the reviewer’s comment, we additionally reported the performance of [8] (CALD) in Figure 3. It can be seen that our method is still competitive or performs better than these baselines, confirming the advantage of the proposed ideas. Including the recent paper [8] of CVPR 2022, we currently believe that we have considered most of the latest active learning techniques for object detection.
> >
> > [6] Yoo, Donggeun, and In So Kweon. "Learning loss for active learning." In Proceedings of the IEEE/CVF conference on computer vision and pattern recognition, pp. 93-102. 2019.
> >
> > [7] Kim, K., Park, D., Kim, K.I. and Chun, S.Y., 2021. Task-aware variational adversarial active learning. In Proceedings of the IEEE/CVF Conference on Computer Vision and Pattern Recognition (pp. 8166-8175).
> >
> > [8] Yu, Weiping, Sijie Zhu, Taojiannan Yang, and Chen Chen. "Consistency-based active learning for object detection." In Proceedings of the IEEE/CVF Conference on Computer Vision and Pattern Recognition, pp. 3951-3960. 2022.
> >
> >
> >
> > Again, thank you for your time and efforts in reviewing our paper. Overall,  we clarified why ReLU performs lower than random baseline, added discussions for handling label noise, and considered another baseline published in CVPR 2022. We hope that these are sufficient grounds for the reviewer to reconsider the rating. We would appreciate further opportunities to answer any remaining concerns you might have.

---

### Author Response · Authors · 2022-11-28
**General comments to AC and all Reviewers**

We appreciate all reviewers for acknowledging the novelty/originality of our work. We also appreciate the reviewers for providing constructive comments, which have greatly helped us to improve the paper. We believe that we successfully addressed every single question/comment, as responded to each reviewer. Below, we provide some major modifications we made during the rebuttal period.

**1. Additional baselines:** Reviewers boP4 and tNAG, updated in Figure 3 of manuscript.

**2. Discussions on how to handle label noise:** Reviewer boP4, updated in Section J of Appendix.

**3. Applicability to two-stage detectors:** Reviewer tNAG, updated in Section N of Appendix.

**4. Discussions on modeling epistemic uncertainty with both classification/regression heads:** Reviewers P2kB and HUy5.

**5. Further experiments on HUA:** Reviewers 4FYw and HUy5, updated in Sections M, L, K of Appendix.

For other details, please refer to our responses corresponding to each reviewer. In case there are remaining questions/concerns, it would be grateful if we can have an opportunity to further answer them.

Best, Authors

---

### Decision · Program_Chairs · 2023-01-20

**Decision:**

Accept: poster

**Justification For Why Not Higher Score:**

The paper is still on the borderline even after reviewers increased their scores. Reviewers still have various minor concerns about this work, motivation why it works well with the label noise etc.

**Justification For Why Not Lower Score:**

N/A

**Metareview: Summary, Strengths And Weaknesses:**

Five reviewers have evaluated this paper and after rebuttal they all scored paper 6 (which his an improvement over initial scores). On that basis, AC advocates an acceptance as a poster and asks authors to include all key aspects of rebuttal into the final paper.

Regarding uncertainty, AC encourages authors to also extend a bit related works as this has been a highly popular topic in recent years. Some ideas include:
A. What uncertainties do we need in bayesian deep learning for computer vision? Kendall et al., NeurIPS'17
B. Towards a Robust Differentiable Architecture Search under Label Noise, Simon et al., WACV'22.
C. Uncertainty-Guided Probabilistic Transformer for Complex Action Recognition, Guo et al, CVPR'22
D. Uncertainty-DTW for Time Series and Sequences, Wang et al, ECCV'22.

For example, [A] contains a very good discussion on what uncertainties are needed in computer vision, [B] uses reparametrization trick/uncertainty modeling for the label noise, [C] uses uncertainty modeling in action recognition while [D] uses reparametrization trick and  epistemic uncertainty in transformers, etc.

AC encourages authors to have a thorough look around to connect to more works on uncertainty and include them into related works.


**Note From Pc:**

if the above contains the word "oral" or "spotlight" please see: "oral" presentation means -> notable-top-5% and "spotlight" means -> notable-top-25%. As stated in our emails, we are disassociating presentation type from AC recommendations